# The Role of Posterior Neural Plate-Derived Presomitic Mesoderm (PSM) in Trunk and Tail Muscle Formation and Axis Elongation

**DOI:** 10.3390/cells12091313

**Published:** 2023-05-04

**Authors:** Barbara K. Stepien, Verena Pawolski, Marc-Christoph Wagner, Thomas Kurth, Mirko H. H. Schmidt, Hans-Henning Epperlein

**Affiliations:** 1Institute of Anatomy, Medical Faculty Carl Gustav Carus, Technische Universität Dresden School of Medicine, 01062 Dresden, Germany; 2Center for Molecular and Cellular Bioengineering (CMCB), Technology Platform, Electron Microscopy and Histology Facility, Technische Universität Dresden, 01062 Dresden, Germany

**Keywords:** DiI injections, GFP^+^ tissue grafting, posterior neural plate, epidermis, presomitic mesoderm (PSM), lateral plate mesoderm, somite, endoderm, axial elongation, axolotl

## Abstract

Elongation of the posterior body axis is distinct from that of the anterior trunk and head. Early drivers of posterior elongation are the neural plate/tube and notochord, later followed by the presomitic mesoderm (PSM), together with the neural tube and notochord. In axolotl, posterior neural plate-derived PSM is pushed posteriorly by convergence and extension of the neural plate. The PSM does not go through the blastopore but turns anteriorly to join the gastrulated paraxial mesoderm. To gain a deeper understanding of the process of axial elongation, a detailed characterization of PSM morphogenesis, which precedes somite formation, and of other tissues (such as the epidermis, lateral plate mesoderm and endoderm) is needed. We investigated these issues with specific tissue labelling techniques (DiI injections and GFP^+^ tissue grafting) in combination with optical tissue clearing and 3D reconstructions. We defined a spatiotemporal order of PSM morphogenesis that is characterized by changes in collective cell behaviour. The PSM forms a cohesive tissue strand and largely retains this cohesiveness even after epidermis removal. We show that during embryogenesis, the PSM, as well as the lateral plate and endoderm move anteriorly, while the net movement of the axis is posterior.

## 1. Introduction

The formation of germ layers and the progressive growth of the embryo in an anteroposterior direction are two crucial steps during early development and morphogenesis of chordates [1,2]. This process occurs in two distinct morphogenetic steps [3]. In the first step, the head and most of the trunk in the anterior part of the embryo are formed, with a central notochord and somites positioned laterally, developing from the gastrulated axial and paraxial mesoderm, respectively [1,2,4,5]. Somites are formed progressively in an anteroposterior direction from an initially unsegmented paraxial mesoderm (therefore denoted as presomitic mesoderm—PSM). In the second step, the embryo grows posteriorly with the addition of prospective posterior trunk and tail somites and the extension of the notochord and neural tube (NT) [2,5]. Although, in certain taxa, the elongation of the posterior body has been described as a continuation of gastrulation, there are many features that make it distinct [3,6]. As an example, most of the endoderm, as well as the intermediate and lateral plate mesoderm (LPM), are only present in the trunk but absent from the prospective tail, while the axial organs (notochord and NT) and PSM occur both in the trunk and tail [7,8]. Critically, the interruption of the formation of the posterior body, e.g., by mutations of Brachyury (T), Wnt, FGF, BMP or RA pathway components, leads to a truncated rostro-caudal elongation of the embryo that largely affects the morphogenesis of the anterior trunk and head [3,6], which is also known as ‘sirenomelia’ [9]. Nonetheless, during the elongation of the posterior body, both the somites and the axial structures (notochord and NT) are produced in a way that displays continuity with the anterior regions with respect to a single embryonic axis. The extension of the PSM in the anteroposterior direction is also thought to be the main driver of posterior body elongation in various organisms [10,11,12]. However, it is still not well understood how the continuity of development and a conserved body plan is achieved in organisms with widely different gastrulation types.

The development of the posterior body is dependent on the tail bud region, an embryonic structure thought to act as an organizer, similar to that of the amniote node or the amphibian dorsal blastoporal lip, during primary gastrulation [2,3,13]. The paraxial mesoderm (PSM) of the posterior trunk and tail, as well as the axial neuroectoderm (NT) are derived from a pool of dual-fated neuromesenchymal (NM) progenitors, which display the transcriptional characteristics of both neural and mesenchymal cell lineages [14,15,16,17,18,19,20]. Notably, they exist in a bipotent state characterized by the simultaneous expression of Sox2 and Brachyury genes that specify either a neural or a mesenchymal cell fate, respectively [15,16,17,19,20,21,22,23,24].

In axolotl (*Ambystoma mexicanum*), such NM progenitors originate from the posterior neural plate (NP), which constitutes approximately the caudal-most fifth of this structure [23,24,25,26]. Later in development, NM cells are located in a posterior progenitor zone (PZ), from which they differentiate either into neural tube (NT) neurons or undergo an epithelial–mesenchymal transition (EMT) [27] to produce mesenchymal cells of the PSM. Although the posterior NP is already partially pre-patterned early on into neuronally (Sox2^high^) and mesenchymally (T^high^) biased zones [22,23], considerable mixing persists, consistent with the notion of a dynamic NM progenitor heterogeneity [21]. Interestingly, a truncated *Xenopus* Xbra mutant that has lost the mesodermalizing activity can lead to the formation of neural structures in animal cap explants [28]. In axolotl, the posterior NP-derived mesenchyme generates one-third of the trunk somites (the posterior trunk region) and all tail somites [23]. During posterior elongation of the axolotl embryo, mesenchymal cells from the PZ migrate posteriorly and then undergo an anterior turn to form two PSM streams, each on one side of the centrally positioned notochord and NT. This mode of PSM morphogenesis during posterior body elongation is unique to the urodeles (tailed amphibians), as opposed to another amphibian model organism, *Xenopus*, or blastodisc taxa, such as fish and birds [2,13,16,17,29]. In axolotl, the generation of the posterior trunk and tail mesoderm is not achieved by typical gastrulation movements, but by an anterior turn of PSM derived from the posterior plate. This anterior turn does not go through the blastopore, but above and posterior to it, and underneath an already specified two-layered epidermis. After the turn, the posterior PSM extends anteriorly, connects seamlessly with the gastrulated mesoderm in the posterior trunk and undergoes segmentation into somites [3,23,25,30,31]. Strikingly, in spite of taxon-specific differences in morphogenesis, similarly structured bodies are produced with a continuous, elongated and anteroposteriorly-oriented axis. A more detailed characterization of the tissue mechanics and single cell behaviours during this process in various species is therefore needed to better understand the developmental principles.

Several studies have described the aspects of PSM tissue movements and the behaviour of individual cells during posterior body morphogenesis [11,12,14,23,32,33]. The overall tissue flow from the dorsal and posterior PZ regions in an anterior direction leads to an anteroposterior formation of somites via PSM condensation. Although this mechanism seems to be conserved [1,19,34], important differences exist between taxa. For example, the posterior elongation in zebrafish is driven primarily by non-volumetric tissue deformation without a major contribution of cell proliferation. This contrasts with observations in mice, in which NM progenitors divide continuously and produce an increase in the overall tissue volume (volumetric growth) [22,33]. Regional gradients of tissue stiffness, cell density and mobility, as well as the role of ECM flow were also postulated to direct the anteroposterior elongation [12,14,35,36,37]. However, the resemblance of these scenarios to the mechanism of PSM morphogenesis in the axolotl has not yet been studied. Moreover, little is known about the contribution of tissues other than PSM and the centrally positioned notochord and NT in axis elongation.

Therefore, in this study, we characterize the morphogenesis of the axolotl PSM and its involvement in axial elongation together with the role of other tissues (epidermis, LPM or endoderm). For this purpose, we used already established techniques of tissue labelling: posterior NP tissue grafting from GFP^+^ donor embryos [23,24] and DiI labelling [38]. Both labelling methods allowed us to assess the relocation of various tissues (paraxial mesoderm, epidermis, LPM or endoderm) from different positions in the developing embryo during the axial elongation of the trunk and tail regions. Tissue grafting was also combined with recently improved optical tissue clearing methods [39,40]. The specific labelling of PSM progenitors (or their subsets), enabled by transparent tissue, permitted the imaging and the 3D reconstructions of developing axolotl tail buds. The parameters describing the morphology of the reconstructed tissue and the behavior of single labelled cells were comprehensively analysed at various developmental stages to create a model of PSM morphogenesis and somite formation in axolotl.

Herein, we show that the posterior elongation of the body in axolotl occurs in two phases: first, as an isovolumetric extension of the PSM caused by tissue reshaping, followed by volumetric growth, which also involves cell proliferation. The tissue generated in the tail bud region of the embryo is shifted anteriorly during development and gives rise to both the somites and part of the NT of the posterior trunk and tail, without the involvement of more anterior trunk regions. The elongation of both the trunk and tail occurs with an anterior shift of all the investigated tissues (except for the epidermis, which remains at the level of the original GFP^+^ grafting site). Moreover, various areas of the posterior NP differ in their spatiotemporal contribution to the PSM and to somite formation. Analysis of individual cell behaviour also reveals changes in the cell shape and orientation, as well as in cell mixing and density, depending on the anteroposterior location of the PSM cells between PZ, various PSM regions and the site of forming somites. Our study also reveals a crucial role of the overlying epidermis as a lateral limit to PSM cell spreading. Taken together, we propose a river-like model of PSM morphogenesis, in which the PSM tissue flow is directed by the adjacent epidermis laterally and the NT/notochord medially. In this model, the migrating mesenchymal cells undergo changes in collective cell behaviour in a spatiotemporal pattern, dictated in part by their place of origin in the posterior NP.

## 2. Materials and Methods

### 2.1. Animals

Embryos of Mexican axolotl (*Ambystoma mexicanum*) white mutant (*d*/*d*) and transgenic GFP-expressing white mutant (*d*/*d*) (*CAGGs:EGFP*; designated as GFP^+^) lines were obtained from the axolotl-facility at the Center for Regenerative Therapies (CRTD), TU Dresden. Eggs of lifeact-GFP transgenic axolotls (*tgSceI(CAGGs:lifeAct)^PMX^*) were kindly provided by Prayag Murawala, Ph.D. (MDI Biological Laboratory, Bar Harbor, ME, USA). For development, the eggs were kept in tap water at room temperature (RT) or 11 °C. The embryos were used for DiI injections, grafting experiments and staining (for a detailed description, see Appendix A).

### 2.2. DiI-Labelling of Tissues

For DiI injections, the embryos (stage 22) were injected with DiI (D3911, Invitrogen) solution in 1 µm/µL in DMSO (D8418, Sigma Aldrich) with the help of a microinjector (3-000-204, Nanoject II, Drummond, Broomall, PA, USA) DiI (5 nl; 1 μg/μL) under a stereomicroscope. After the injections, the embryos were kept separately in an agar-coated dish at RT or 11 °C and imaged daily until stage 40. For each injected embryo, the ‘relocation’s relative distance’ (d_rel_ = a/b) was calculated, where a is the distance between the anterior-most end of the animal and the center of the labelled region in parallel to the ventral body lining of the embryo, and b is the total length of the animal. For a detailed description of the dye injection procedure, see Appendix A.

### 2.3. Homotopic Grafting of GFP^+^ Tissues

For tissue grafting, the donor and host embryos were operated on in 4 × Steinberg’s solution [41] at 11 °C. The GFP^+^ tissue was grafted from the GFP^+^ donors to white mutant (*d*/*d*) hosts (homotypic, isochronic grafts). The embryos were then allowed to develop until an appropriate stage. The exact experimental procedures are described in the Appendix A.

### 2.4. Histological Stainings and Optical Tissue Clearing

Vibratome sections were used for immunostaining with chicken α-GFP (Abcam, ab13970) 1:1000 and rabbit α-Sox2 [42] 1:500. Whole-mount staining for 3D reconstructions was performed according to a procedure adapted from Kurth and colleagues [43,44]. Whole-mount staining for plastic sections (immunostaining and toluidine blue staining) was performed according to [23,43]. For optical tissue clearing, either of the two methods was applied: ethyl cinnamate-based optical tissue clearing, as in [40], for whole-mounts or SeeDB optical tissue clearing [39] for sections. For details of the staining, tissue clearing and imaging procedures, see the Appendix A.

### 2.5. Image Processing and Analysis

The 3D reconstructions were either produced with Fiji [45] or Arivis Vision 4D 3.3.0 (Arivis AG) software. Manual cell segmentation, filopodia segmentation and GFP^+^ tissue volume calculations were obtained using Fiji 3DManager [46] and 3D Viewer Plugin [47]. Arivis was used for automatic cell segmentation of the GFP^+^ cells and the generation of custom adjusted transversal and sagittal planes, fitting to the anatomy of individual embryos. For the details of tissue morphogenesis and individual cell parameter analysis, see the Appendix A.

## 3. Results

### 3.1. Morpohological Changes during Axial Elongation of a Developing Axolotl Embryo

To facilitate a detailed investigation of the morphogenetic movements of the posterior NP-derived PSM, we first analysed the shape changes occurring in the axolotl embryo between a neurula (stage 14) and a prehatching larva (stage 40; Figure 1) based on the data from [48]. As these descriptions are too short, we must provide more details here. The stages from a fertilized egg to a late gastrula have a constant diameter of 2 mm and are spherical. From an early neurula (stage 14; Figure 1a) onwards, embryos become elliptical with a progressively longer anteroposterior (AP) axis. The neural folds become elevated, approach each other towards the dorsal midline (middle neurula, stage 16; Figure 1b) and fuse (stage 19; Figure 1c). At stage 22, an early tail bud stage has developed with fused neural folds and a prospective head, trunk and tail (Figure 1d,n,o). The embryo elongates along the AP axis through stages 28 (Figure 1e) and 30 (Figure 1f). At stage 35, a larva with a stretched AP axis and an extended tail is formed (not shown) that continues to lengthen while the head, trunk and tail structures become more clearly discernible. The pre-hatched larva (stage 40; Figure 1g) has an elongated body with a central NT and a notochord in the trunk and tail, flanked bilaterally by paraxial mesoderm, and is covered with epidermis (Figure 1j–m). The tail differs from the trunk because it lacks the LPM and endoderm [7,8].

The development of the axolotl embryo between an egg and a larva at approximately stage 43 proceeds externally. It relies on the cellular yolk supply until the onset of predation. Hatching and the first food intake occur by stage 43 [49]. Therefore, the body elongation during these phases could be predicted to be largely non-volumetric, i.e., based on tissue remodelling without an increase in the overall volume [33]. To approximate such volume changes, we analysed the dimensions of axolotl embryos reported in [48] from the egg until stage 40. The embryo remains spherical with an average diameter of 2–2.1 mm until stage 14 (early neurula), when it starts to elongate (Figure 1h,i). From that stage on, the length of the embryo along the AP axis increases steadily, reaching 9.3 mm at stage 40. This is accompanied by a decrease in embryonic width (mediolateral dimension) between stages 15 and 24 (2.1 to 1.35 mm). The height of the embryo (dorsoventral dimension) increases between stages 20 and 22 (1.7 to 2.3 mm), and then varies up and down, ranging between 1.6 (stages 33–35) and 2.1 mm (at stage 40). During the later stages, the width and height do not change significantly, despite the increase in the length of the embryo and its number of somites, as previously described [48]. This suggests that during late pre-larval and early larval stages, the embryo grows volumetrically. The source of such growth during the pre-hatching stages in the absence of external feeding remains unclear. The tissues involved in the formation and elongation of the embryonic axis are the spinal cord and notochord as axial organs, and the paraxial mesoderm left and right to them.

### 3.2. Differential Contribution of Embryonic Tissues to Axial Elongation

First, we analysed the behaviour of different embryonic tissues derived from the three germ layers during the elongation of axolotl embryos. For this purpose, DiI injections (Figure 2, Appendix A) or GFP^+^ tissue grafting (Appendix A) techniques were used. To uncover the differences in the behaviour of various trunk and tail regions, the tissues were analysed in the following manner. A number of small areas in the dorsolateral, midbody or ventral positions along the embryonic AP axis were labelled at stage 22. The embryos were then allowed to develop until stage 40. The displacement of the DiI- or GFP-labelled tissue regions relative to the anterior end of the animal, as well as the localization of labelled cells in different embryonic structures was assessed (Figure 2 and Appendix A).

#### 3.2.1. Displacement and Tissue Contribution of Labelled Dorsolateral Paraxial Mesoderm

DiI was injected into 5 different dorsolateral regions of the paraxial mesoderm along the AP axis of the trunk at stage 22 (each injected region in a separate cohort of animals). With the injections, either segmented mesoderm (somites) in the head–trunk transition zone (Figure 2a, purple) and mid-trunk (Figure 2a, orange), or presomitic mesoderm (PSM) in the trunk–tail transition zone (Figure 2a, green), tail bud (Figure 2a, blue) and tail tip (Figure 2a, red) were focally labelled. The head–trunk transition zone is localized posteriorly to the brain bulge. The mid-trunk position is the dorsal apex of the embryonic trunk, and the trunk–tail transition zone is defined by an imaginary transversal line through the prospective cloaca (for injection sites see Figure 2a). For the number of DiI injections, see the legend of Figure 2. The position of the labelled cell populations was tracked over the developmental time (stages 30, 35 and 40). Representative images of the labelled embryos at different stages are shown in Figure 2c. Areas with labelled cells from all the injection sites became shifted anteriorly during body elongation (Figure 2b), but the degree of the anterior shift varied, with the largest anterior displacement occurring in the trunk–tail transition site (Figure 2a,b, green). The smallest shift occurred when DiI was injected at the head–trunk transition zone (Figure 2a,b, purple). This shows that anterior head growth is largely unaffected by the elongation of paraxial mesoderm regions located more posteriorly. Interestingly, the cells from the posterior-most injection site in the tail tip (Figure 2a,c) became localized both in the neural tissue of the tail tip and in the mesoderm of the tail and posterior trunk. This distribution is consistent with the existence of dual-fated NM progenitors at the posterior-most tail tip of stage 22 axolotl embryos. This is also the reason why injections into the tail bud were labelled only tail NT but not mesoderm.

#### 3.2.2. Displacement and Tissue Contribution of Labelled LPM and Endoderm of the Lateral Trunk and Tail Bud

DiI was injected into three different lateral body sites along the AP axis of stage 22 embryos (each injected region in a separate cohort of animals). The injections were labelled LPM and, occasionally, endoderm (the outer lining of the future mucosa and yolk platelets) of the lateral trunk (Figure 1n and Appendix A, orange and green) and tail bud (Appendix A, red) regions. The future smooth muscles of the gut derived from the visceral layer of the LPM were also occasionally labelled. A double labelling of the LPM and endoderm could not always be avoided due to the thin LPM (about 30 µm), which made precise targeting of this tissue challenging (see Figure 1n,o). The injected regions are schematically shown (Appendix A). The position of the labelled cell populations was tracked over time (stages 30, 35 and 40). Representative images of the labelled embryos at different stages are shown in Appendix A. From stage 22 to stage 40, all the labelled tissues shifted anteriorly (Appendix A). Lateral injections at the mid-trunk position showed a large expansion of the labelled area (Appendix A, orange and c, top panel). In addition to an anterior shift, the labelled cells also spread ventrally, likely due to penetration of DiI into the endoderm. Tissue from the trunk–tail injection site showed a less extensive expansion accompanied by ventral spreading (Appendix A, green and c, middle panel). None of the labelled tissues from these injection sites could be detected in the posterior trunk or tail at stage 40. In contrast, the cells from injection sites in the tail bud region were labelled both posterior trunk myotomes and tail spinal cord, reminiscent of dorsal paraxial mesoderm injections at this AP axial level (compare Figure 2). As there is no LPM in the tail bud, the injections were likely labelled paraxial tail mesoderm in addition to distalmost endoderm. These results suggest that the LPM and endoderm do not contribute to the extension of the posterior axis.

#### 3.2.3. Displacement and Tissue Contribution of Labelled Endoderm of the Ventral Trunk

DiI was injected into three different ventrally-located regions along the AP axis of stage 22 embryos (each injected region in a separate cohort of animals). The injections reached the endoderm (the outer lining of the future mucosa and, occasionally, yolk platelets; see Figure 1o) of the ventral trunk (Appendix A, purple and orange) and the trunk–tail transition zone close to the prospective cloaca (Appendix A, green). The LPM layer is not yet present ventrally at this stage (Figure 1o). The injected regions are shown schematically (Appendix A). The position of the labelled cell populations was tracked over time (stages 30, 35 and 40). Representative images of the labelled embryos at different stages are shown in Appendix A. In contrast to the dorsolateral injection sites, the labelled tissue did not show significant displacement until stage 30 (Appendix A). From stage 35 onwards, the labelled tissue from all the injection sites shifted anteriorly. The largest anterior shift could be measured for the injection sites at the trunk–tail transition zone. Here, the labelled cells were present not only in the endoderm of the hindgut, but also in the somites and somite-derived ventral fin mesenchyme, which is consistent with the somite origin of the fin mesenchyme [24]. These were the only labelled cells found posteriorly to the cloaca. As with lateral injections into this region, the labelling of the somites must have been caused by having co-labelled endoderm and PSM of the tail bud erroneously, due to close tissue proximity. These results suggest that the ventral endoderm does not contribute to either the extension nor shrinkage of the axis.

#### 3.2.4. Displacement and Tissue Contribution of the GFP^+^ Epidermis and Paraxial Mesoderm of the Dorsal Trunk and Tail

The relative movement of the epidermis and underlying paraxial mesoderm is of interest with respect to differential shifts in the trunk and tail tissues. DiI injections are not suitable for specific labelling of the epidermis due to the difficulty in precise targeting and possible dye diffusion into deeper tissue layers. Therefore, grafting experiments with GFP-labelled tissues were used instead (Appendix A). Tissue from GFP^+^ donors to white (*d*/*d*) hosts was grafted at four different regions along the dorsolateral trunk of stage 22 embryos (Appendix A): mid-trunk, trunk–tail transition zone, the tail bud and the tail tip. For each region, either the surface epidermis or the underlying paraxial mesoderm (somite or PSM) was transplanted (each transplanted region in a separate cohort of animals) to allow a direct comparison with DiI-labelled dorsal somite/PSM regions (3.2.1). The results of the GFP^+^ labelling of the paraxial mesoderm were similar to those obtained with the DiI injections. They showed a progressive anterior displacement of the grafted tissue (Appendix A,h,j). In comparison, the GFP^+^ epidermal grafts essentially stayed in their initial locations (Appendix A,g,i). Next, the relative displacement of the epidermis- and paraxial mesoderm-graft derived tissues at stage 40 were evaluated and compared (Appendix A). In all the regions, the labelled mesoderm had shifted more anteriorly than the overlying epidermis. Only the posterior tail PSM transplants contributed to the labelling of a broad area containing myotomes of the posterior trunk and tail (Appendix A), in agreement with the results of the DiI labelling (3.2.1). The green fluorescent ventral region in Appendix A is an artefact due to yolk autofluorescence.

To conclude, the combination of the DiI injections and GFP^+^ tissue grafting experiments allowed us to study the behaviour of different tissues (epidermis, paraxial mesoderm, LPM and endoderm) during axis elongation. All of these tissues except for the epidermis show a clear anterior displacement (Figure 2b, Appendix A). Only the paraxial mesoderm, but not the LPM or endoderm, which are absent from the developing tail region (Figure 1l,m), contributes to posterior axis elongation. This is in agreement with previous studies in axolotl [23,24], in which the posterior trunk and all tail somites, as well as parts of the posterior NT and the dorsal and ventral tail fins, are derived from the PSM of the tail bud (Figure 2 and Appendix A). An anterior displacement of the DiI-labelled tissues, despite an anteroposterior growth of the embryonic axis, may be surprising. This is, however, consistent with the notion that the embryonic tail does not expand posteriorly simply by adding tissue to the posterior end of the animal.

### 3.3. Dual Cell Fate of the Posterior NP Progenitors (Grafting of the Entire GFP^+^ Posterior Neural Plate)

The posterior NP, which can be defined as the posterior-most fifth of the neural plate in the axolotl neurula at stage 15, was shown to contain the prospective PSM progenitor population [23,24]. In the first part of this section, we used vibratome sections to identify the cell fate of GFP^+^-labelled grafts from the entire posterior fifth of the neural plate (hereupon designated as posterior neural plate). The GFP^+^ posterior plate grafted to a white host gives rise to the entire somite material of the tail and about one-third of somites in the posterior trunk [24]. In addition, a small portion of the GFP^+^ cells differentiate into neurons of the early tail bud and this dual fate can be explained by the existence of a prospective mesoderm (Bra^+^) and neuroectoderm (Sox2^+^) progenitor overlap zone [23,24], likely corresponding to dual-fated neuromesenchymal progenitors as described for other taxa [14,15,16,17,18,20,21]. The developing PSM (unsegmented paraxial mesoderm) undergoes a progressive anterior-to-posterior segmentation into somites [1,2,34]. To investigate the course of development of posterior NP-derived PSM into somites, we transplanted the GFP^+^ posterior NP to *d*/*d* hosts at stage 15 and followed the morphogenesis of the PSM through stages 22, 28 and 30 (GFP^+^ somites present) on paramedian sections (Figure 3a,b). Due to the lack of a mesenchyme-specific marker, which makes the labelling of the PSM cells in the axolotl impossible, we combined GFP staining (which marks all cells derived from the posterior NP graft) with Sox2 neuronal staining, in addition to DAPI staining of the nuclei. This allows for identification of the PSM cells as GFP^+^ and Sox2^−^. At stage 22, immunostaining with GFP and Sox2 (Figure 3b; image adapted from [23]) shows that GFP^+^ Sox2^−^ cells were present posteriorly in the PZ bulge. They were followed more anteriorly by a mixture of GFP^+^/Sox2^−^ cells and only Sox2^+^ cells in the posterior tube (Figure 3b). At stage 28, a widespread GFP staining of the PZ and the PSM bulge, with a few spread-out GFP^+^ PSM cells present anteriorly, was detected in whole mount GFP/Sox2 double-stained embryos. NT tissue was predominantly Sox2-positive and ended in single GFP^+^ cells (Figure 3c). Therefore, the PSM must have been derived from the posterior NP to a large extent. A few individual GFP^+^ cells contributed to somite formation (Figure 3c, arrows) intermixed with unlabelled host-derived cells. This intermixing underscores the continuity of trunk and tail somitogenesis. At stage 30, two to eight pairs of somites largely consisting of GFP^+^ cells were formed, and one pair of somites was added every 2 h (Figure 3d). Overall, the axolotl tail bud region contains NM progenitors destined for NT or PSM, as described in other species [16].

To obtain a holistic picture of PSM morphogenesis, in addition to studying GFP^+^ PSM development on vibratome sections and whole mounts as above (Figure 3b–d), we decided to also analyse this system in 3D tissue reconstructions of the tail region in whole mounts of axolotl embryos. In these reconstructions, individual GFP^+^ cells were segmented and analysed (Figure 3e). Optical tissue clearing based on ethyl cinnamate [40] combined with GFP immunostaining allowed for confocal microscope imaging through the entire thickness of the tissue from the left aspect of the animal to the midline (signals to the right of the midline become too weak due to increased light scattering). As proof of principle, examples of such reconstructions at stages 19, 22, 28 and 30 are shown in Figure 3e. Graft-derived cells were immunostained against GFP and the NT was pseudo-coloured red. At stage 19 (early tail bud), the GFP^+^ tissue accumulated in a dorsomedial area of the caudal tail bud. This tissue mass remained localized there through stages 22 and 28, while the GFP^+^ cells became depleted from this area by stage 30. A bilateral PSM stream started to form ventrally from the PZ at stage 22 and extended progressively in an anterior direction, showing the anterior turn behaviour [23]. By stage 28, the PSM extended along the medially located NT and notochord, nearly doubling in length (see Figure 4c). The anterior-most GFP^+^ PSM cells were mixed with unlabelled host trunk PSM cells and formed somites. By stage 30, the somites derived from the anterior region of the labelled PSM were predominantly GFP^+^. In contrast, the GFP^+^ cells of the posterior PSM region were sparse. Throughout the tail bud, a small number of GFP^+^ cells were also detected in the NT, consistent with the contribution of some posterior NP progenitors to the spinal cord neurons [23]. The 3D reconstructions (Figure 3e) show a tissue architecture of the tail bud that is comparable to that of the vibratome sections (Figure 3b) [23]. This offers a technical possibility to assess the morphological changes in tissue dimensions and individual cell behaviours of large cell groups during PSM development.

### 3.4. D Reconstructions of the PSM

We used the established 3D reconstruction method (3.3) to analyse PSM morphogenesis in detail. In the first part of Section 3.4.1, we grafted the left half of the posterior NP to investigate the changes in regionalization and PSM tissue dimensions during development, including contralateral migration of PSM cells. In the second part (3.4.2.), we grafted small median, paramedian or lateral regions of the left posterior neural plate in order to study subregionalism of the PSM strand.

#### 3.4.1. Three-dimensional Reconstructions of PSM Grafts of the Left Half of the GFP^+^ Posterior NP

To obtain detailed knowledge of the spatiotemporal changes in PSM size and morphology, we first grafted the left half of posterior NP from a GFP^+^ donor to a white (*d*/*d*) host embryo at stage 15 (Figure 4a). Unilateral grafting allowed the tracking of cells that crossed the midline so that contralateral spreading of the cells within the developing PSM could be assessed. Except for the analysis of the midline crossing, we only considered the 3D-reconstructed PSM on the left side of the midline. Chimeric embryos were analysed at stages 22, 28 and 30. In this time period, the GFP^+^ PSM continually increased its length from around 500 to 1200 µm (Figure 4b,c). Although a small number of GFP^+^ cells were already incorporated into the trunk somites at stage 28, the majority of GFP^+^ somites first appeared at stage 30 (Figure 4b). Most of the GFP^+^ tissue between stages 22 and 28 formed the PSM. Between stages 22 and 28, the labelled PSM volume remained constant and doubled only between stages 28 and 30 (Figure 4d). This shows that the elongation of the axolotl PSM in an anteroposterior direction occurs in two phases: first, as tissue reshaping without overall volume changes (stages 22–28), and then by volumetric growth (stages 28–30) (Figure 4c,d). This switch between non-volumetric morphogenesis and volumetric growth is reminiscent of the changes in overall embryo dimensions approximating body volume during development (Figure 1h,i). This may appear surprising because axolotl embryos develop externally, and the larvae do not consume food until about stage 43 [49]. Instead, they consume yolk platelets, as reflected by the measured volume of the individual cells, which becomes reduced by half between PZ at stage 22 and all PSM regions, as well as newly formed somites throughout stages 28 and 30 (Figure 4e). This is in contrast to the hypothesis proposed by [33], which claims that an external mode of development is associated with non-volumetric growth. This discrepancy can, however, be resolved, if the increase in PSM volume is caused by mechanisms that do not require external sources of energy, such as an intake of fluid into the deposited ECM.

The changes in PSM elongation are accompanied by dynamic shape changes of its height and width along the AP axis. At stage 22, the GFP^+^ mesoderm forms a bulge in the tail bud (Figure 4b). At stage 28, this bulge decreases in width and forms a PSM that is extended in height and length (Figure 4b,f,g). At stage 30, this PSM becomes more elongated with decreasingly smaller areas with GFP^+^ cells at its anterior end where somites are forming. Here, GFP^+^ PSM is mixed with unlabelled cells. At this stage, all the areas of the GFP^+^ tail mesoderm had a regular height of about 140 µm. The GFP^+^ mesoderm of the left posterior plate became fully integrated into the paraxial mesoderm of the left trunk. Under brightfield illumination, no morphological differences were visible between the operated side containing the GFP^+^ mesoderm and the unlabeled contralateral side.

#### 3.4.2. Three-dimensional Reconstructions of PSM from Grafts of Median, Paramedian or Lateral Subregions of the GFP^+^ Posterior NP

Grafting the entire or only the left half of the GFP^+^ posterior NP has shown that the graft-derived tissue conducts a coherent morphogenetic movement within the host. However, how the local behaviour of small groups of cells contributes to this large-scale movement is unclear. NM progenitors destined for the PSM were shown to undergo a high degree of cell mixing within the progenitor zone in other species [12,32]. In order to assess whether an analogous cell mixing occurs in axolotl or if the cells form coherent subpopulations with similar migratory behaviour, small GFP^+^ grafts were transplanted from a median (Figure 5a), paramedian (Figure 5b) or lateral (Figure 5c) position of the left posterior NP into white (*d*/*d*) hosts at stage 15. In this way, local patterns of cell migration can be studied from the starting position in the posterior NP until the final location in a somite/myotome. Each transplant position was analysed at stages 28, 30 and 35. From all the transplant positions, only the median posterior NP GFP^+^ cells became distributed to the contralateral side of the NT (Figure 5a). The GFP^+^ cells grafted into a paramedian (Figure 5b) or a lateral position (Figure 5c) mostly remained on the ipsilateral side with only a small number of cells crossing the midline in the PZ region at stage 30. However, these cells did not contribute to contralateral PSM or somites at later stages. Ipsilaterally, at stage 28, the GFP^+^ cells grafted in the median position had undergone the anterior turn and reached the medial and anterior PSM (for region definitions, see Appendix A, 8.4), but had not yet reached the somite positions (Figure 5a). At stage 30, these cells almost left the median PSM and had contributed to the anterior PSM and the formation of somites, followed, by the formation of myotomes at stage 35. None of these cells were found remaining in the PZ at the end of the tail bud. Meanwhile, the GFP^+^ cells derived from grafts into a paramedian position (Figure 5b) at stage 28 were found along the entire length of the PSM from its posterior region up to the anterior PSM. Few cells were also lagging in the PZ region. These cells also contributed to somites and myotomes by stage 30 and 35, respectively. The GFP^+^ cells grafted into a lateral position were still accumulated in the PZ and posterior PSM regions at stage 28 (Figure 5c). At stage 30, these cells either remained in the PZ (3 embryos) or had moved anteriorly (4 embryos). The anterior reach of the cells grafted laterally resembled that of paramedian grafted cells; however, they were shifted more ventrally and laterally, away from the NT (Figure 5c). At stage 35, these cells migrated further anteriorly and participated in myotome formation. Thus, the migratory behaviour of the posterior NP graft-derived GFP^+^ differs depending on their original mediolateral position. The medial-most cells exit the PZ and spread anteriorly earlier and reach more anterior locations than cells from the paramedian and lateral positions.

We summarize the characteristics of the PSM tissue flow at different developmental stages in Table 1 below. Taken together, we show that the axolotl PSM forms a continuous and cohesive tissue strand (Figure 4, Table 1), which flows from the dorsocaudal PZ bilaterally to the notochord and NT in an anterior direction, where it undergoes segmentation into somites. Despite seemingly disparate origins and tissue mechanics during the formation of anterior and posterior body somites [3], there is a remarkable continuity of the process of anteroposterior axis extension and somitogenesis between these body parts. In 3D reconstructions, posterior neural plate-derived GFP^+^ cells intermix seamlessly with unlabeled host cells in more anterior trunk regions that originate from the gastrulated mesoderm (Figure 3, Figure 4 and Figure 5). We observed somites made nearly exclusively from the gastrulated host PSM with only a few GFP^+^ cells, as well as somites consisting entirely of non-gastrulated GFP^+^ posterior NP cells (Figure 3c,d). No uncontrolled lateral or medial deviations of the GFP^+^ PSM cells were observed, and even single anterior GFP^+^ cells become integrated into the host PSM or into somites. Furthermore, we demonstrated that a complete compact PSM strand (here, a left one) may be described as being composed of PSM substreams that originate from different posterior NP regions in a distinct spatiotemporal pattern (Figure 5, Table 1). We could observe remarkable migratory behaviour of these PSM tissue substreams, with each of them displaying different timing of migration. Moreover, only cells from the median position showed very limited crossing to the contralateral side. These results demonstrate a striking spatiotemporal order of the PSM tissue flow.

**Table 1 cells-12-01313-t001:** Analysis of collective behaviour and individual cell parameters obtained in 3D reconstructions of PSM tissue strands during PSM morphogenesis. Abbreviations: PZ: posterior zone; post.: posterior; med.: medial; ant.: anterior. ^1^ cell shapes in PZ: oval; in post. PSM: elongated; in med. PSM: oval; in ant. PSM: oval; in somite: elongated; NA: not applicable.

Parameter	Stage 19 (69 h)	Stage 22 (73 h)	Stage 28 (92 h)	Stage 30 (102 h)	Stage 35 (122 h)
GFP+ PSM stream	NA	tail bud PSM bulge	Extended PSM	Elongated PSM, anteriorly narrowed	NA
GFP+ PSM stream (from median post. NP)	NA	NA	Medially from medial to anterior PSM	Medially in anterior PSM and somites	Medially in myotomes
GFP+ PSM stream (from paramedian post. NP)	NA	NA	Medially from posterior to anterior PSM	Medially from posterior PSM to somites	Medially in myotomes
GFP+ PSM stream (from lateral post. NP)	NA	NA	PZ and posterior PSM	Laterally from posterior PSM to somites	Laterally from PSM to myotome
GFP+ PSM cell number	527	1095	1232	1989	NA
AP-oriented cell divisions [%]	5.9	1.4	12.1	4.4	NA
Cell orientation	NA	PZ: 48% mediolateral	Post. PSM: mediolateral and anteroposterior (36% each); med. PSM: 44% anteroposterior	Ant. PSM: 32% mediolateral, 30% anteroposterior; somites: 46% dorsoventral, 33% anteroposterior	NA
Cell distances (cell group cohesion)	NA	PZ: 14.9 µm	Post. PSM: 14.3 µm; med. PSM: 57.9 µm	Ant. PSM: 10.6 µm; somites: 10.2 µm	NA
Cell distances (density)	NA	NA	NA	PZ to ant. PSM: ~30 µm; somite 1: ~25 µm	NA
Cell shape (aspect ratio; see Figure 6c-d); for description see ^1^	NA	PZ: 2.33	Post. PSM: 2.78; med. PSM: 2.26	Ant. PSM: 2.37, somite: 3.03	NA
Filopodia length	NA	PZ: ~6 µm	Post. and med. PSM ~6 µm	Ant. PSM and somites ~4 µm	NA

**Figure 6 cells-12-01313-f006:**
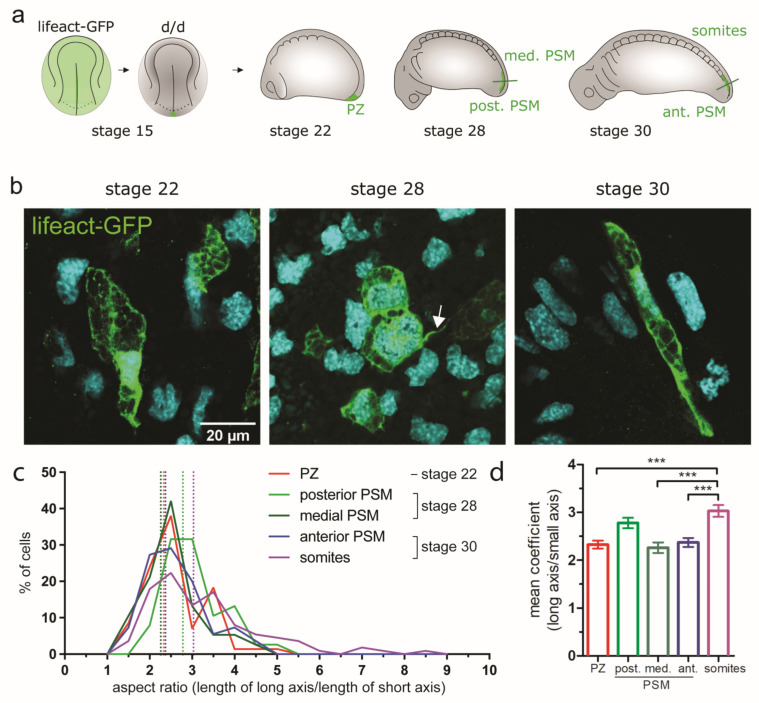
Changes of cell shapes in the PSM. (**a**) Experimental design: a small number of cells from the medial posterior NP was transplanted from a stage 15 lifeact-GFP^+^ donor to a *d*/*d* host. (**b**) Examples of lifeact-GFP labelled cells within the PSM at stages 22, 28 and 30; single planes of optically cleared vibratome sections; for axial planes of sectioning, see (**a**). A filopodium is indicated with an arrow. (**c**) Cell shape distribution per region. Cellular coefficients (length aspect ratios of long axis/short axis) represent cell shape. (**d**) Bar plot of the mean cellular coefficient corresponding to the data in (c). Error bars indicate SEM (upper whisker). Significant differences are indicated (1-way ANOVA: *** *p* < 0.0001). Number of analysed cells (per region): *n* = 71 (PZ), 38 (post. PSM), 38 (med. PSM), 55 (ant. PSM) and 112 (somites). Number of embryos: *n* = 3–4 embryos per stage.

### 3.5. Analysis of Cellular Parameters Influencing Paraxial Mesoderm Morphogenesis in the Posterior Body

To understand individual cell contributions to the observed large-scale and local tissue behaviour during PSM morphogenesis, multiple cellular parameters relating to cell division and migration were analysed (summarized in Table 1).

#### 3.5.1. Cell Addition and Oriented Cell Divisions of Tail Tissues during Morphogenesis and Elongation

To understand the factors contributing to the volumetric and non-volumetric length increases of the developing axolotl tail region (compare Figure 4), we assessed the changes in the GFP^+^ cell number in embryos after posterior NP transplantation at stage 15. The GFP^+^ cells were counted in immunostained transverse sections from chimeric embryos at stages 19, 22, 28 and 30 (Table 2). The numbers from all sections where GFP^+^ cells could be detected were added up. Mesodermal and neuronal cells were distinguished based on Sox2 staining. The vast majority of cells at all the analysed stages (100–94%) were presumably mesodermal cells (Sox2^−^) (Table 2). The number of GFP^+^ graft-derived mesodermal cells roughly doubled between stages 19 and 22, but only a modest increase in these cell numbers occurred between stages 22 and 28 (around 10%). Interestingly, the latter period corresponds to the time of non-volumetric elongation of the PSM (Figure 4c,d), suggesting cell division has no significant role in this process. Subsequently, the number of GFP^+^ cells increases by around 60% between stages 28 and 30 (Table 2). Curiously, this period of increase in cell number corresponds to the period of volumetric growth of the PSM (Figure 4c,d), which argues for the role of cell proliferation during this phase. We have not assessed the effect of apoptosis or other types of cell death; however, previous studies in mouse PSM suggest that apoptosis in this tissue is limited and unlikely to have a major effect on morphogenesis [50].

Directional tissue growth can also be a result of oriented cell divisions, which plays a role in processes such as the expansion of the neuroepithelium [51]. To test if oriented cell divisions play a role in anteroposterior PSM expansion, individual mitotic cells were identified based on the nuclear DNA appearance in thick vibratome sections stained with propidium iodide (for a representative image, see Appendix A). The positions of the centrioles in mitotic cells were marked and the orientation of a mitotic spindle was determined for each individual cell with reference to the AP axis. Embryos at stages 19, 22, 28 and 30 were analysed and the results are summarized in Table 2. The percentage of anteroposteriorly-oriented mitotic spindles was low and ranged from 1.4% to 12.1%. Thus, oriented cell division does not appear to be the main driver of PSM extension along the AP axis.

In summary, the increase of the PSM cell number correlates temporally with the increase in volumetric, but not non-volumetric growth (Table 1 and Table 2). However, the directional posteriorly-directed PSM expansion seems to be driven neither by defined proliferation centers, nor by a bias towards AP-oriented cell divisions.

#### 3.5.2. Changes of Cell Shape and Orientation during Tail Morphogenesis and Elongation

Tissue morphogenesis can also be mediated by changes of cell volume, shape and orientation. To assess whether any of these parameters play a significant role in the elongation of the PSM, individual cells within unlabelled tissue were selectively labelled. This allowed for easy segmentation and quantification of single cells. For these experiments, small posterior NP grafts comprising around 20 cells were transplanted from a medial location in lifeact-GFP transgenic donors to white (*d*/*d*) axolotl hosts at stage 15 (Figure 6a). In this axolotl line, GFP is genetically linked to actin fibres, which allows the visualization of the cytoskeleton, including cytoplasmic protrusions (Figure 6b). Chimeric embryos were analysed at stages 22, 28 and 30. To visualize the labelled cells, the embryos were optically cleared with the SeeDB clearing protocol [39], which allows a lower tissue imaging depth than the ethyl cinnamate-based protocol but is compatible with actin staining. To accommodate this lower access to deep-lying tissues, the cell morphologies were analysed in 80 µm transverse vibratome sections rather than in whole-mount 3D reconstructions.

First, the shapes of the labelled PSM cells were analysed. The cells were manually segmented to calculate the length/width aspect ratios. At stage 22, lifeact-GFP^+^ cells were located exclusively in the PZ. By stage 28, the transplanted cells were present mostly in the posterior and medial PSM and, at stage 30, in the anterior PSM and somites. In the graft-derived cells, GFP-labelled actin fibres were localized at the cell borders, between the yolk platelets inside the cells and in filopodia (Figure 6b). Lifeact-GFP-labelled cells residing in different regions (PZ; posterior, medial and anterior PSM; and somites) showed a wide morphological variety, from round (length/width aspect ratio close to 1) through elliptical (aspect ratio > 1 and < 5) to highly elongated (aspect ratio > 5) (Figure 6c). Examples of such cells can be seen in Figure 6b. The distribution of the calculated length/width aspect ratios (Figure 6c) and their mean coefficient (Figure 6d) were similar across the different regions, with the exception of somites, which contained more elongated cells, as expected from previous studies [30,31,34]. All the cells in the PZ and PSM regions had an aspect ratio ranging between 1.5 and 4.5–5, with a peak at 2.5, with the exception of the posterior PSM, for which the aspect ratio of most of the cells ranged from 2 to 5 (with a broad peak at 2.5–3). The aspect ratios for the somite cells started at 1.5 with a peak at 2.5, but cells with an aspect ratio up to 8.5 could also be found, in accordance with the presence of a subset of cells that are more elongated than in the other areas. The mean coefficients of the aspect ratios were calculated for the cells in all regions. The mean coefficients for the cells in the PZ, medial and anterior PSM were 2.33 (SD 0.7), 2.26 (SD 0.68) and 2.37 (SD 0.71), respectively. They were significantly lower than the mean coefficient of somites (3.03; SD 1.31). The posterior PSM cells had a mean coefficient of 2.78 (SD 0.67), with a trend intermediate between PZ and other PSM regions on one side and somite cells on the other. In summary, in the developing PSM, elongated cells are more frequent in the posterior PSM than in other PSM regions. Most of the elongated cells could be found in the forming somites. These were previously reported to be highly elongated [30,31,34]. This is unlikely to be associated with motility, but rather reflects cell shape change concomitant with epithelization as a result of the mesenchyme-to-epithelial transition [34]. In contrast, the elongation of the PSM cells before somite formation could be associated with their higher motility during tissue morphogenesis. The cells also became elongated when entering the posterior PSM, but rounded up again when reaching the medial to anterior PSM. This result is in agreement with earlier SEM findings, according to which anterior PSM cells in axolotl embryos become more rounded before forming somites [30,31].

Cell orientation can indicate the direction of cell migration, as well as influence the shape of tissues. We assessed whether the preferred cell shape changes in various regions of the developing posterior trunk and tail mesoderm were associated with a bias for the defined cell orientation. For this purpose, we re-analysed the previously segmented lifeact-GFP^+^ cells (Appendix A). We quantified cell orientations in the PZ, PSM and somite regions (Appendix A). In the PZ, labelled cells were oriented primarily mediolaterally (48%) (Appendix A). In the PSM regions, the cells became oriented more anteroposteriorly, with a maximum in the medial PSM (44%). In the anterior PSM, the cells rounded up and became randomly oriented before they participated in somite formation. In the somites, the cells were strictly elongated and oriented predominantly dorsoventrally (46%). Concurrently, 33% of the somite cells were oriented anteroposteriorly (Appendix A) (see also [23]), consistent with the rosette morphology [30,31]. The results of the study on cell shape and orientation are summarized in Table 1 and graphically in Appendix A.

The uncovered changes in cell shape and orientation between developing PSM regions in axolotl embryos are likely correlated with the speed and direction of cell migration. Particularly, the increased elongation and preferred anteroposterior orientation of posterior PSM cells suggest their participation in the anterior turn movement, originating from posterior NP cells. Many of the lifeact-GFP^+^ cells in the PSM studied in optically cleared vibratome sections showed filopodia (Figure 6b, arrow). In contrast to the PZ and somites, the majority of cells in all the PSM regions had at least one filopodium (Appendix A). A small subset of cells in all the regions had five or more filopodia, but their proportion increased markedly in the anterior PSM. The length of the filopodia ranged from 1 up to 39 µm (Appendix A). The average length, however, was close to 6 µm in the cells from the PZ to medial PSM regions. The anterior PSM had, on average, more but shorter filopodia (average length of 4 µm) and the filopodia remained similarly short in the somites.

Taken together, we demonstrate that PSM cells change their cell shape and orientation during anterior migration (Figure 6 and Appendix A). At stage 22, PZ cells are located in the tail bud, are moderately elongated and oriented mediolaterally (summarized in Table 1). These cells move into the posterior and medial PSM by stage 28. First, they become elongated and gradually change their orientation into an anteroposterior direction while migrating anteriorly. In the medial and anterior PSM, they become less elongated again, before assuming a very elongated shape and a primarily dorsoventral or anteroposterior orientation in the nascent somite. This is consistent with the previous reports of cells becoming elongated as they organize in rosette-like structures in the forming somites [30,31,34].

#### 3.5.3. Cell Group Cohesion during Axial Elongation and PSM Morphogenesis

Neighbouring cells in developing tissue can display a different group cohesion (a tendency for a local group of cells to stay in close proximity over time) depending on their motility. While cells that do not migrate stay close together, migrating cells can show different behaviour. Cell mobility can lead to an increase of distances between individual cells and thus lower group cohesion, when the direction and speed of movement of individual cells are random and uncorrelated. In contrast, directed cell migration associated with highly correlated cell mobility (similar direction and speed) maintains high group cohesion. To assess the group cohesion in migrating PSM cells, the graft-derived lifeact-GFP^+^ cells (see 3.5.2) were re-analysed. We assessed the distances between the individual labelled cells of the transplanted group in different regions of the PSM (Figure 7). Here, the ellipsoid-fitting method used for the analysis of cell orientation (Appendix A) was utilized to calculate a central mathematical point for each ellipsoid (designated as the center of a cell). To quantify the cohesion of a transplanted cell group, the pairwise distances between the center points of each cell in a pair of labelled cells in each tissue section were calculated (Appendix A).

In the PZ at stage 22, the measured relative cell distances were low, with a median distance of 14.9 µm and the majority of cells remaining within 25 µm of each other (Figure 7b,g). As the cells moved to the posterior PSM (Figure 7c,g), the median distances between the individual cells remained low (median distance 14.3 µm) and a large proportion of the labelled cells still maintained close proximity (< 25 µm distance). This suggests a highly correlated migration of individual cells in the posterior PSM. Upon further migration into the medial PSM region, however, a nearly equal number of long and short cellular distances was measured, with a median distance of 57.9 µm, almost 4 times higher than in the PZ (Figure 7d,g). This suggests a loss of cell group cohesion and a low coordination of Individual cell movements. Interestingly, the relative cell distances decreased significantly in the anterior PSM (median distance 10.6 µm) (Figure 7e,g) and remained low in the somites (median distance 10.2 µm) (Figure 7f,g). These values are even lower than the distances between the labelled cells in the PZ (Figure 7b,g). In summary, the measurements of relative cell distances (Table 1, cell distances (cell group cohesion)) support a model of coordinated cell migration in the PZ and posterior PSM, which switches to more random movements in the medial PSM, leading to extensive cell mixing.

### 3.6. The Epidermis Governs Tail Elongation by Affecting Mesodermal Tissue Integrity

Recently, a posterior body elongation model has been proposed, which predicts a mechanical feedback loop between tissue compression and expansion [10]. In this model, anteriorly migrating PSM cells compress the centrally positioned notochord and NT from both sides. This creates pressure on these axial organs, which causes their posterior expansion, and, in turn, a push is generated, which induces tissue flow from the PZ into the PSM. This model requires the existence of a lateral border to inhibit PSM cell spreading and maintain tissue pressure. The manner in which this lateral limit is provided and whether its maintenance is indeed necessary for axial elongation is still unclear. Given that the epidermis lies directly laterally to the developing PSM (Figure 1j–m), we decided to test the hypothesis of its role in limiting the spreading of PSM cells.

For this purpose, chimeric embryos were produced by grafting the left GFP^+^ posterior neural plate to white (*d*/*d*) host embryos at stage 15 (Appendix A). The embryos were allowed to develop to stage 22, at which point the epidermis was removed uni- (on either left or right side) or bilaterally from the tail bud posterior to the last-formed somite. Chimeric embryos with the left GFP^+^ posterior neural plate grafts without epidermis removal were used as controls. At stage 30, the morphology of the embryos was assessed for tail malformations. In addition, the embryos were fixed, immunostained with α-GFP and α -Sox2 antibodies, optically cleared and imaged. The images were then used to create 3D reconstructions for a more detailed analysis of tissue malformations. The removal of the epidermis changed the morphology of the tail in the majority of the operated embryos (Table 3), whereas only a minority of the control embryos displayed a missing tail (likely due to the effect of the transplantation procedure), but no tail bending or other severe malformation phenotypes. Representative examples of control and epidermis-deprived embryos are shown in Appendix A. Lateral tail bending always occurred towards the side where the epidermis was removed. A misshapen tail was often bent dorsally and accompanied by an exposed posterior endoderm/prospective cloacal region, which shifted dorsolaterally. Bilateral epidermis removal also increased the frequency of embryos missing an entire tail region (Table 3).

Subsequently, 3D reconstructions were used to assess the tail malformation at the level of tissue morphology. Tail bending in epidermis-deprived embryos was accompanied by a combined bending of the NT and PSM tissue into the same direction. At the same time, the PSM had shortened or expanded ventrally. However, the measurements of the GFP^+^ tissue height, width or length did not reveal significant differences in tissue dimensions upon epidermis removal (Appendix A). Similarly, there was no significant change in the GFP^+^ tissue volume (Appendix A). These results can be explained, at least in part, by a wide variability of observed malformations in individual embryos compared to relatively low morphological variation between the control embryos (Appendix A).

To understand whether epidermis removal may affect tissue spreading at a single cell level, the ‘closest neighbour analysis’ was used. GFP-labelled cells in the chimeric embryos were segmented and grouped inside a sphere according to their regional localization (Appendix A). The distances to the closest neighbours were calculated for the cells within each group as a measure of cell density. As the GFP^+^ cells derived from the left posterior NP transplants were located primarily on the left side with minimal contralateral crossing (Figure 3 and Figure 4), the effects of unilateral epidermis removal on the behaviour of the cells on the ipsi- (left) or contralateral (right) side could be assessed (Appendix A). In addition, the effects of bilateral epidermis removal were quantified (Appendix A). In an unoperated control embryo, the cell density between the PZ and anterior PSM remained stable (Appendix A, Table 1, cell distances (density)). At stage 30, the cells in the anterior PSM, in front of the somite formation, could not migrate any further, therefore the cells in the somites become densely packed (Appendix A). In contrast, a significant increase in the relative cell distances was observed with both bilateral and ipsilateral epidermis removal (but not contralateral removal) within the PZ and posterior PSM regions. The removal of epidermis on the contralateral side was accompanied by a decrease of the cell distances. This suggests that the epidermis does not only function as a passive barrier to lateral cell spreading but may have an active role in stimulating and directing cell movements by secreting ECM components [52]. In contrast, epidermis removal had no effect on the cell density in the anterior PSM. The relative cell distances in the first three posterior-most somites (with S1 being closest to the tail tip) were affected to a different degree, depending on the side of epidermis removal. In all cases, the significant changes led to decreased cell density in the affected somite, which could likely be explained by cells lagging in the posterior PSM due to lateral spreading.

To summarize, we show that the integrity of the PSM strand is nearly unaffected by the removal of the epidermis, which serves as a direct lateral boundary for the developing paraxial and LPM (Figure 1j–o). This result is in contrast, for example, to epidermis removal in the head of axolotl embryos [53], after which cranial neural crest-derived branchial arches forming collective migratory streams lose their tissue coherence and become flattened. Nonetheless, an ipsilateral removal of the epidermis from the posterior region of an axolotl embryo at stage 22 exerts minor but significant effects on PSM compaction and thereby on tail morphogenesis (Appendix A).

The dorsolateral epidermis is known to contribute ECM components, particularly fibronectin (FN), to the underlying PSM [54] and the extracellular material flow has been shown to direct randomly moving cells within this tissue [12]. Therefore, we conducted tail formation rescue experiments to distinguish between the importance of the epidermis in providing a lateral physical barrier to PSM tissue spreading versus an active function in secreting ECM components to the PSM. The epidermis was removed on the left side of embryos at stage 22 and the embryos were embedded in either soft 3% methylcellulose or a stiffer 1% agarose gel to manipulate the stiffness of the physical barrier (Appendix A). Stage 22 embryos in which no epidermis removal was performed were embedded in the same conditions to serve as controls. The methylcellulose embedding had no effect on the development of the unoperated control embryos, but the more rigid agarose affected the extension of the embryos in the anteroposterior direction. Therefore, the agarose embedding was removed from the anterior and posterior embryonic regions, leaving its limiting effect to the lateral, dorsal and ventral aspects of the embryo. In addition, both the embedding techniques were combined with the addition of the ECM component fibronectin to mimic the matrix deposition role of epidermis on cell migration [54,55]. As shown in Table 4, neither 3% methylcellulose nor 1% agarose embedding could rescue the tail development defects, indicating that the physical barrier function of the epidermis alone is not sufficient to correct the defects of PSM morphogenesis. An addition of FN to the methylcellulose also failed to rescue the defects. Interestingly, however, the combination of the stiff physical barrier provided by 1% agarose with FN led to rescue in 2 of the 3 embryos in which the epidermis was removed (one of the control animals also showed a tail formation defect). The representative images of the control and epidermis-removed embryos embedded in different conditions are provided in Appendix A. Taken together, these results suggest a dual role of the epidermis, both as a physical barrier to PSM cell spreading and as a provider of a permissive ECM.

## 4. Discussion

In this study, we analysed the morphogenesis of the axolotl (*Ambystoma mexicanum*) posterior NP-derived PSM and its role in somite formation and axis elongation, in addition to several other tissue types (epidermis, LPM and endoderm). In summary, we have investigated the following: (1) the displacement of various embryonic tissues enveloping the embryo by using focal DiI injections and GFP^+^ tissue grafts (Figure 2, Appendix A); (2) the morphogenesis of the PSM tissue strand, including its subregions, by using GFP^+^ tissue grafting and 3D reconstructions (Table 1); (3) the behaviour of individual GFP-labelled PSM cells and small cell groups during PSM morphogenesis (Table 1); and (4) the role of the epidermis in maintaining the coherence of the PSM tissue strand (Table 3 and Table 4). Based on these results, and in the context of previous studies, we proposed a model of PSM morphogenesis and axis elongation in axolotl.

### 4.1. Model of PSM Morphogenesis and Axis Elongation in Axolotl

Based on the analysis of PSM morphogenesis in the developing axolotl embryo (summarized in Table 1) presented here, we propose a model of PSM formation and morphogenetic movements, which contributes to axis elongation in the posterior body (Figure 8). We propose that the morphogenesis of the axolotl PSM corresponds to a ‘river-like flow’ model, which involves cooperation between an internal PSM strand cohesion (fluid-like tissue) and an external epidermis (lateral limit of the PSM, acting like riverbanks). The PSM forms a coherent tissue strand, akin to a dense fluid, in which the cells ‘flow’ from the PZ anteriorly to the position of somite formation. The cells in this tissue strand do not mix randomly, but are instead well-organized in spatiotemporally distinct substreams, which originate from the median, paramedian and lateral regions of the posterior NP that can be distinguished experimentally via grafting experiments. Thus, median posterior NP cells lead a stream of cells bilaterally to the NT in a polonaise manner, followed by slower cells exiting from more lateral regions. This model of PSM cell flow resembles a river, where the fastest flow speed occurs in the middle of the stream, with decreasing speeds towards the lateral boundaries. These boundaries are generated by the epidermis, which acts as a stiff lateral barrier. It also modulates tissue flow by producing ECM components, such as FN [54] (Table 4), which permit cell movement and guide cell migration [35,52,56,57]. The loss of epidermis leads to an increase of the distance between the PSM cells, consistent with a higher occurrence of lateral PSM tissue spreading (Appendix A). Crucially, even such small differences in lateral spreading can have profound effects on tail development, leading to deformations or a complete loss of a tail (Appendix A, Table 3). Overall, the PSM tissue strand is elongated posteriorly by a posterior-to-anterior directed addition of the PSM mesenchyme. The net displacement and growth of the PSM occurs in the posterior direction, while the overlying epidermis stays in the same position. Thereby, the PSM cells are directed anteriorly towards the site of somite formation, while the PSM narrows and the embryo elongates posteriorly.

This model of PSM morphogenesis is consistent with a mechanical feedback loop between compression and expansion proposed in chicken [10,36], where left and right PSM cells migrating anteriorly compress the NT and notochord on either side. Such pressure induces an expansion of the axial organs into a posterior direction and thus pushes the mesodermal cells from the PZ into the PSM. One could envision two different mechanisms that shape the cohesion and flow of the PSM: (1) internal cohesion, which depends on reciprocal contacts and the coordination of cell movements, or (2) an externally-mediated tissue compaction imposed by external restrictions on the spreading of PSM tissue. The second mechanism was hypothesized to provide boundaries to the lateral expansion of the PSM in chicken development [12], but its existence, to the best of our knowledge, has not been demonstrated experimentally until now. We show that a relatively stiff epidermis provides a lateral migration barrier, which may direct the PSM pressure on the axial structures. Axial elongation in chickens has been also suggested to require the combination of opposing anteroposterior gradients of cell motility and cell density [12], and a similar mechanism was proposed in zebrafish [37]. Our results in the axolotl are consistent with these models. Less densely packed and more mobile cells are present in the posterior and median PSM than in the forming somite. This may be understood as a conservation of certain aspects of tissue mechanics during PSM morphogenesis between taxa, although great differences exist with respect to the tissue architecture, which arise during and after gastrulation.

Taken together, our data support a model where coordinated cell migration in the PSM maintains it as a compact tissue strand. Nonetheless, a continuous lateral barrier is necessary, together with a medial one supplied by the NT and notochord, to constrain and direct PSM tissue flow and to allow somite formation.

### 4.2. Limitations and Further Research Directions

Despite the application of the technological advantages of optical clearing and the techniques to reconstruct an entire axolotl PSM tissue in 3D, the main limitation of the current study lies in its use of fixed samples. This unfortunately limits our analysis to certain points in time, which only allow us to deduce live patterns of cell migration based on a number of parameters, such as shape or orientation. Since we observe a high degree of heterogeneity in these parameters, it would be of interest to follow the migration of individual cells live, particularly in relation to the speed of migration of cells from different posterior NP regions. Moreover, whether the cells in these different regions migrate with different speeds or start their migration with a temporal delay remains unclear. The differences in cell mobility, but not in inherent directionality of migration, have been proposed to be a characteristic feature of posterior PSM cells in other model organisms [12,14]. Instead, the direction of cell movement could be primarily directed by the overall tissue flow and/or the mechanical forces provided by the lateral and medial boundaries. Although our results are consistent with such a model, future experiments involving live imaging may allow direct measurements of the speed and direction of single cells and their changes in time and space. Combining live imaging of the PSM with a manipulation of mechanical forces acting on the PSM, e.g., by removal of the epidermis or ablation of axial structures, is needed to refine the model of PSM morphogenesis and axial elongation proposed here.

## 5. Conclusions

What is the main tissue driving axis elongation and tail formation in axolotl? Although we still do not have a definitive answer, we can try to formulate a hypothetical model based on the roles of various tissues investigated herein (the PSM, epidermis, LPM and endoderm) and probable function of other tissues inferred from the literature. While the axolotl gastrula is still spherical (stage 13), a very first anterior–posterior extension occurs by a convergence and extension movement of the neural plate at stages 14 to 16 (Figure 8a). As a consequence, the embryo becomes elongated, and the neural folds raise and fuse in the midline (stages 20–21). At stage 15, the notochord does not even reach the posterior NP and thus cannot yet influence axis elongation. The same applies to the paraxial mesoderm. By stage 15, only 1–2 pairs of somites are present in the anterior part of the trunk, which, therefore, could not cause a posterior extension of the body yet. During stages 19 to 22 (Figure 8b), the posterior NP becomes converted into PSM and reverses its posterior direction of migration into an anterior one in order to connect with the gastrulated mesoderm. Thereby, PSM cells change their shape and orientation and give rise to myotomes (stage 28) (see Figure 8c,e). Meanwhile, the notochord has grown posteriorly. It now has the same length as the NT, with which it is caudally fused by the chordoneural hinge (CNH) linking the floorplate and notochord. Behind the CNH, the posterior wall is located, a stem cell-like region and major source of axial progenitors, with prospective neural (dorsal) and prospective somitic (ventral) fates [19,23]. Until the end of tail formation, the progenitors for NT, notochord and muscle remain in this location [19]. At stage 30, the embryo is still curved, and it is not before stage 35 that, for the first time, a larva with an extended axis and a stretched tail has developed (Figure 8d).

The tail curvature in earlier stages could be due to endogenous physical properties of the NT/notochord and that of a still very soft PSM that does not allow an extension yet. Alternatively, the PSM growth might not match the timing of the posterior growth of the NT/notochord and these two axial organs cannot extend. In any case, a curved tail does not seem to be caused by a time delay brought about by the production of PSM. In *Xenopus*, in which PSM is produced directly and more quickly from gastrulated mesoderm, no time delay occurs, but early embryos still have a curved tail. After the anterior turn of the PSM is finished, the PSM behaviour in axolotl resembles that in *Xenopus* with a gastrulated mesoderm. The youngest PSM is now produced at the posterior end and continuously added to older PSM and somites anteriorly. Thereby, the net movement of PSM and the elongation of the axis constantly moves posteriorly. It is also conceivable that the timepoint of axial extension is coordinated with the production of fin mesenchyme. Dorsal and ventral fin cells are provided by the developing somites rather than the trunk neural crest [24]. However, the somites are likely not able to produce fin tissue before stages 30-31 [49]. Only from these stages onward, the tail becomes gradually stretched and takes up its function. The LPM and endoderm seem to have a negligible role in axial extension and tail elongation. DiI injections of these tissues into selected trunk regions anterior and posterior to an imaginary vertical line through the mid-trunk apex only showed slight anterior displacements of the injection sites. In contrast, as shown by GFP^+^ tissue transplantation, the epidermis seems to stay in the same axial position during elongation, yet, we have shown an indispensable role of the epidermis for tail development. Without it, the tail cannot develop properly. The embryos, from which epidermis was partially posteriorly removed at stage 22 on the left, right or on both sides, showed mild to severe malformations of the tail. To summarize, during axial elongation, the neural plate plays a role very early, followed by cooperation between the extending notochord and the NT and PSM. The PSM and a part of the posterior NT are generated by the PZ and added in a posterior-to-anterior fashion. The PSM flow is directed by the stationary epidermis and the somites become formed in an anteroposterior direction. Finally, by stage 31, the dorsal and ventral fins have developed, and the entire tail is overgrown and stabilized by the epidermis, which enables its stretching. In this way, a functional whole has developed.

## Figures and Tables

**Figure 1 cells-12-01313-f001:**
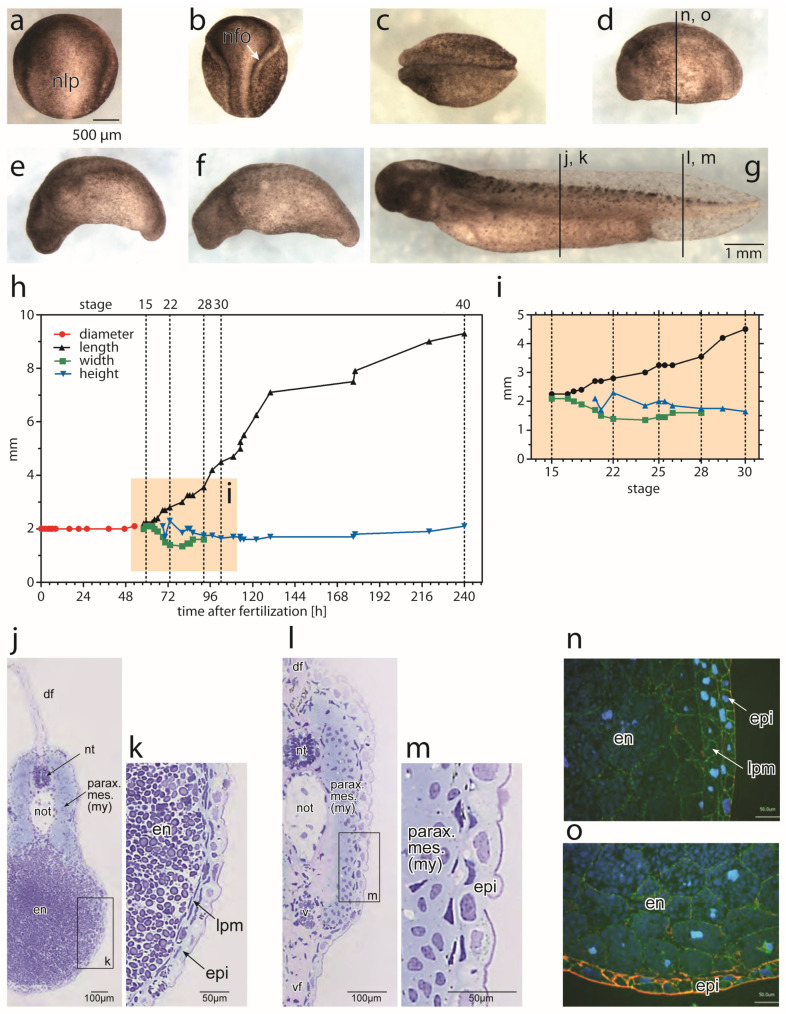
Axis elongation in developing axolotls. Major post-gastrulation stages. (**a**) Neurula, stage 14 (dorsal view, anterior up); (**b**) neurula, stage 16 (dorsal view); (**c**) neurula, stage 19 (dorsal view); (**d**) tail bud, stage 22 (lateral view, anterior left); (**e**) tail bud, stage 28 (lateral view); (**f**), tail bud, stage 30 (lateral view); (**g**) larva, stage 40 (lateral view); staging after [49]. (**h**) Measurement of body dimensions during embryonic growth. (**i**) Insert from (**h**), original measurements after [48]. (**j**–**m**) Toluidine blue-stained transverse sections through the trunk ((**j**), and enlargement, (**k**)) and tail ((**l**), and enlargement, (**m**)) at stage 41, demonstrating axial organs (neural tube: nt; notochord: not), mesodermal components (paraxial mesoderm: parax. mes.; lateral plate mesoderm: lpm), dorsal and ventral fins: df and vf, respectively, and endoderm (future gut): en. For the axial levels of sectioning, see (**g**). Both the paraxial and lateral plate mesoderm are present in the trunk; the latter is absent in the tail. (**n**,**o**) Plastic sections through the mid-trunk of stage 22 wholemounts (for the axial level, see (**d**)) stained with DAPI (blue), α-actin (red) and α-ß-catenin antibodies. The LPM forms a thick layer between the epidermis and the endodermal lining (future mucosa) in the lateral trunk (**n**), but is absent from the ventral trunk at this stage (**o**). Other structures are denoted as npl: NP; nfo: neural fold; v: blood vessel; and my: myotome (denotes paraxial mesoderm after stage 35).

**Figure 2 cells-12-01313-f002:**
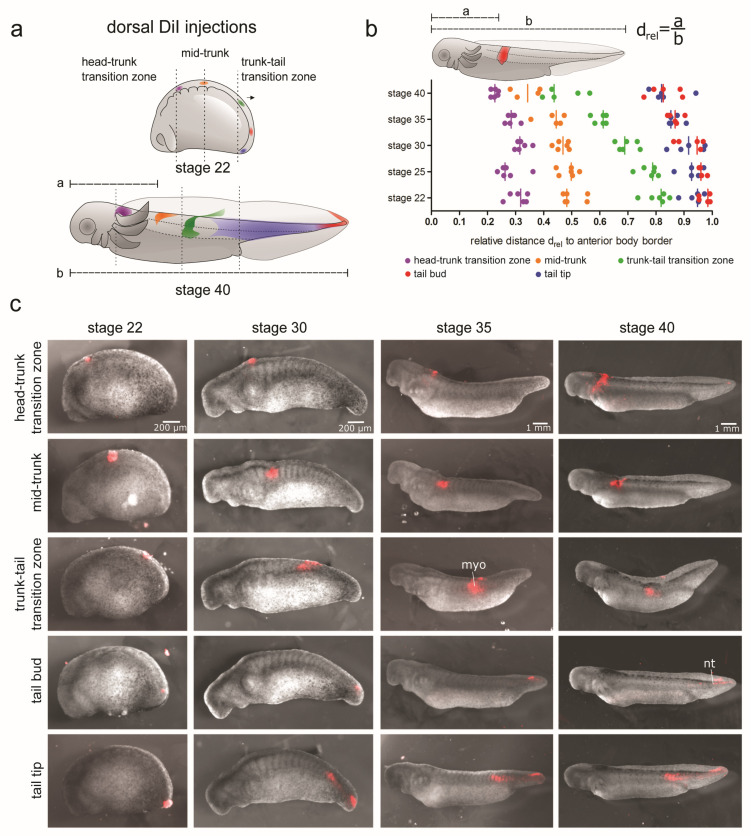
DiI-labelling of paraxial mesoderm in the dorsal trunk and tail of a tail bud (stage 22). (**a**) Schematic displaying focal DiI injection sites at stage 22 and after expansion at stage 40. (**b**) Quantification of DiI label locations at various stages after the injection at stage 22. D_rel_ indicates relative distance to anterior body border. (**c**) Example images of embryos showing distribution of DiI injection sites from the time of injection (stage 22) to a larval stage (stage 40). Number of embryos injected per site: *n* = 6 each for DiI injections into the head–trunk transition zone (violet), mid-trunk (orange) and trunk–tail transition zone (green); *n* = 5 each into an area in the tail bud (red) and tail tip (blue); myo: myotome; nt: neural tube.

**Figure 3 cells-12-01313-f003:**
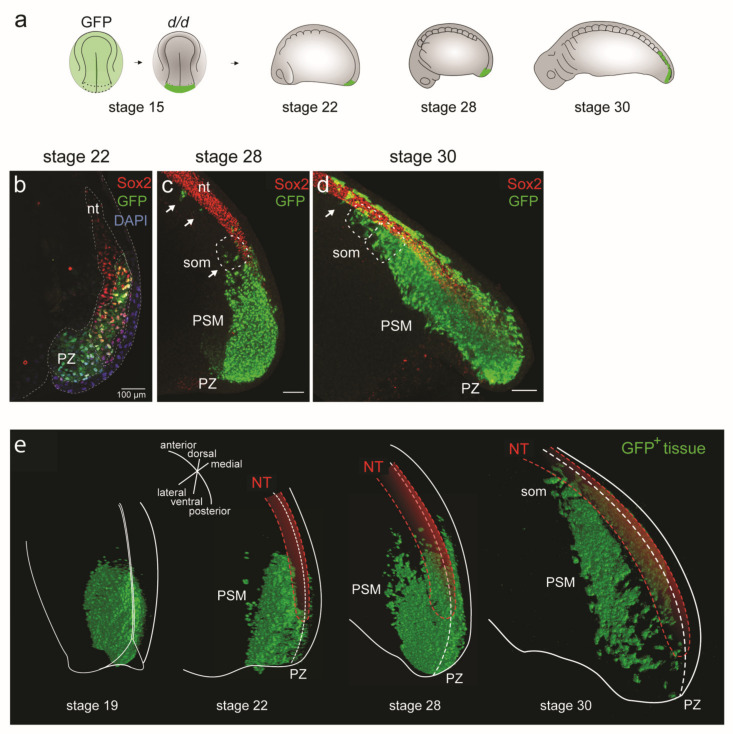
Median histological sections and 3D reconstructions through GFP^+^-stained posterior NP-derived tail tissue at stages 22, 28 and 30 showing PSM/neural development in axolotl embryos. (**a**) Experimental design: The entire posterior NP was transplanted from a stage 15 GFP^+^ donor to a *d*/*d* host; hosts were grown to stages 22, 25 and 30. (**b**) Paramedian vibratome section through PSM at stage 22 stained with DAPI and immunostained against Sox2 and GFP. Image previously published and adapted from [23]. (**c**,**d**) Maximum intensity projections through a z-stack at stages 28 and 30 (immunostaining against Sox2 and GFP with optical clearing). (**e**) Three-dimensional reconstructions at stages 19, 22, 28 and 30 showing distributions of transplanted GFP^+^ posterior NP tissue immunostained against GFP. Posterior parts of tail buds including neural tubes are outlined. Orientation of embryos is outlined in the coordinate diagram in (**e**). NT: neural tube; PZ: posterior zone; PSM: presomitic mesoderm; som: last formed somites. Arrows point to a few individual GFP^+^ cells that contribute to somite formation.

**Figure 4 cells-12-01313-f004:**
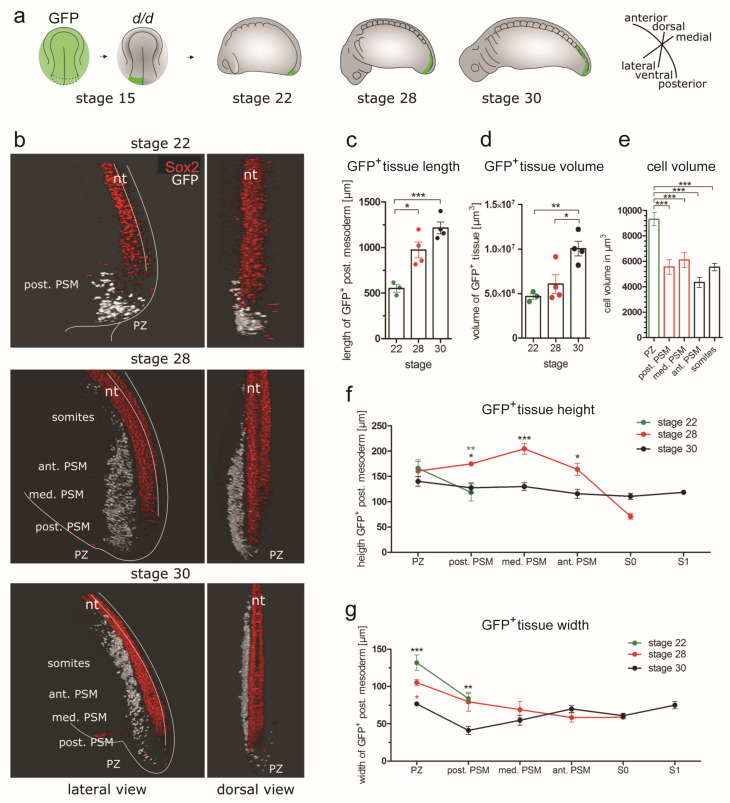
Measurement of PSM tissue dimensions. (**a**) Experimental design: The left side of the posterior NP was transplanted from a stage 15 GFP^+^ donor to a *d*/*d* host. (**b**) Host embryos were immunostained against GFP and Sox2, optically cleared and imaged with a laser scanning microscope at stages 22, 28 and 30. The GFP^+^ tail mesoderm was 3D-reconstructed. (**c**) Length (anterior–posterior extension) of GFP^+^ tissue; (**d**) GFP^+^ tissue volume; (**e**) volume of individual GFP^+^ cells in thick vibratome sections. Number of cells analysed (per region): *n* = 71 (PZ, stage 22), 38 (post. PSM, stage 28), 38 (med. PSM, stage 28), 55 (ant. PSM, stage 30), and 112 (somites, stage 30). (**f**) Height (dorsoventral extension) and (**g**) width (mediolateral extension) of the GFP^+^ tissue in different stages. Significant differences are indicated (1-way ANOVA; * *p* > 0.005, ** *p* > 0.001, *** *p* < 0.0001). Number of embryos analysed per stage: *n* = 3 (stage 22), 4 (stage 28), and 4 (stage 30). Orientation of embryos is outlined in the coordinate diagram. PZ: posterior zone; PSM: presomitic mesoderm with anterior, medial and posterior zones (arbitrarily defined); nt: neural tube.

**Figure 5 cells-12-01313-f005:**
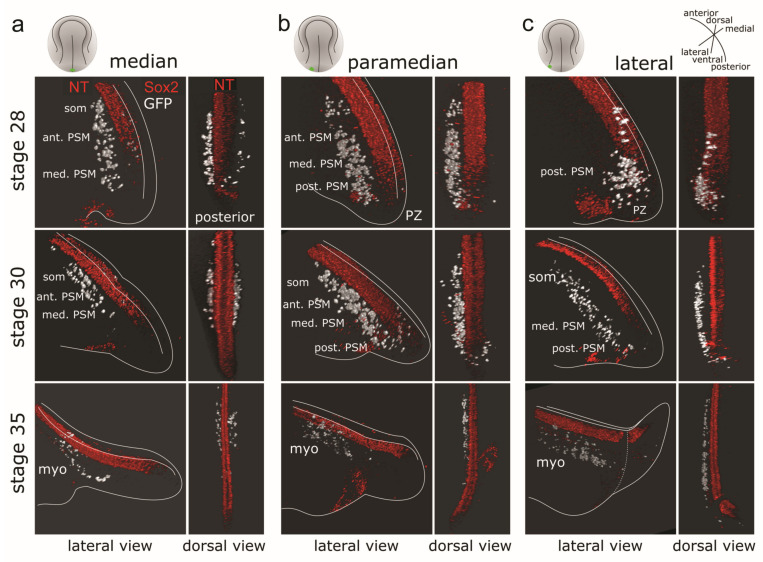
GFP^+^ cells from median, paramedian and lateral regions of the posterior NP become distributed differently during the anterior turn. Small grafts of median (**a**), paramedian (**b**) or lateral (**c**) posterior neural plate tissue were transplanted from a GFP^+^ donor (stage 15) into *d*/*d* hosts and grown to stages 28, 30 and 35. Three-dimensional reconstructions of whole-mount immunostained embryos were optically cleared and imaged. Median posterior NP cells became located more anteriorly than lateral cells. Orientation of embryos is outlined in the coordinate diagram. NT: neural tube; PSM: presomitic mesoderm with anterior, medial and posterior zones, som: somites; myo: myotome.

**Figure 7 cells-12-01313-f007:**
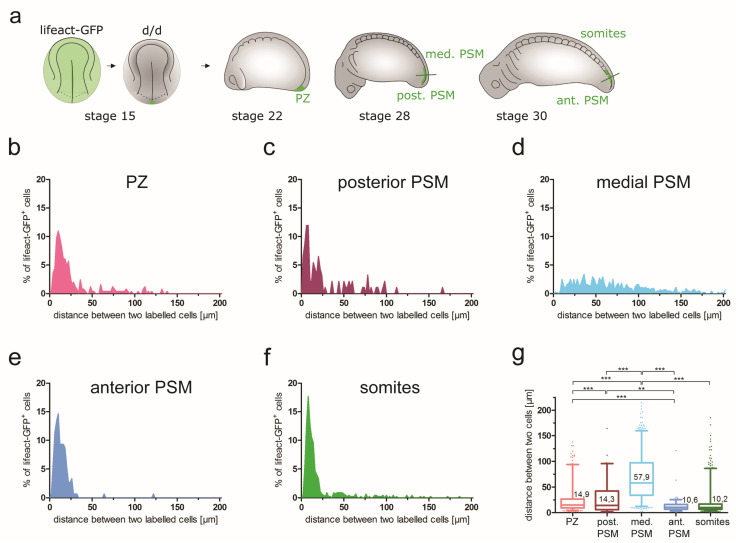
Distances between labelled cells as a measure of cell group cohesion. (**a**) Experimental design: a small portion of the medial posterior neural plate was transplanted from a lifeact-GFP^+^ donor to a *d*/*d* host at stage 15 and chimeric embryos were analysed at stages 22, 28 and 30 using vibratome sections of the labelled regions. (b–f) Plots of the frequency (y axis) of pairwise distances between lifeact-GFP^+^ cells (x axis) in indicated regions: (**b**) PZ; (**c**) posterior PSM; (**d**) medial PSM; (**e**) anterior PSM; and (**f**) somites. (**g**) All measured distances were plotted for each region. The plot shows 5-95 percentiles and medial values. Significant differences between regions are shown (1-way ANOVA: ** *p* < 0.001, *** *p* < 0.0001). Numbers of analysed pairwise distances (per region): *n* = 246 (PZ), 92 (posterior PSM), 437 (medial PSM), 150 (anterior PSM) and 446 (somites). Number of analysed embryos: *n* = 3–4 embryos (per stage).

**Figure 8 cells-12-01313-f008:**
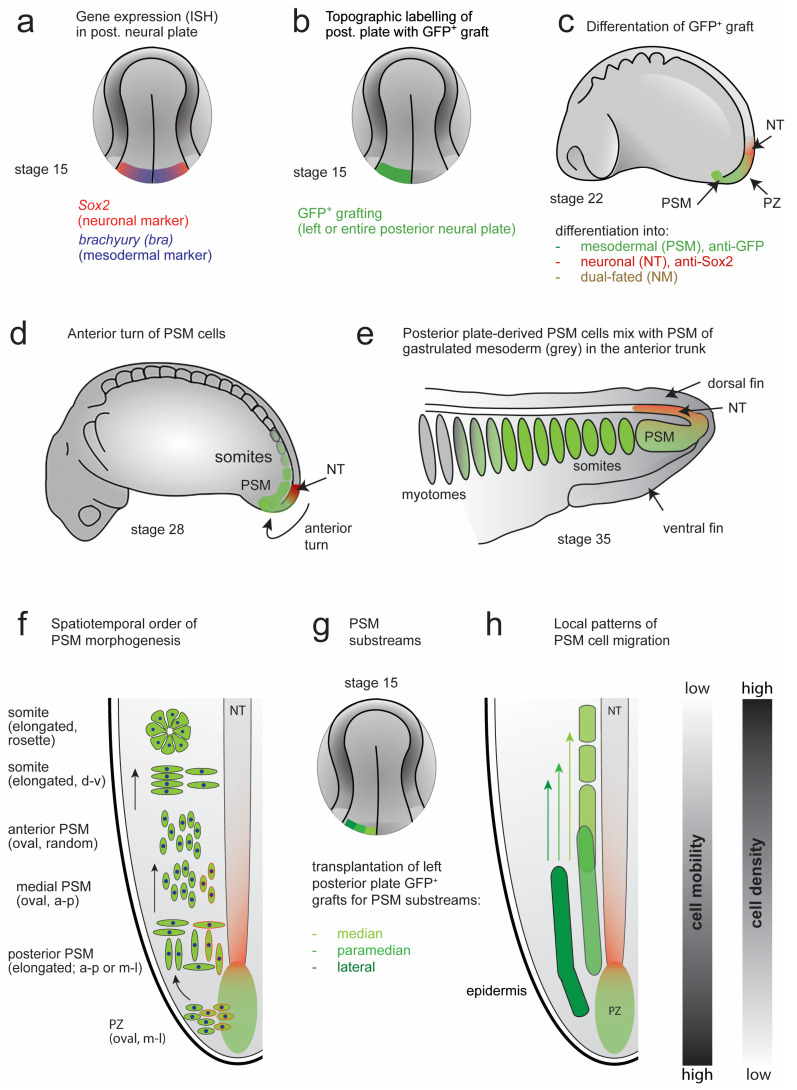
Model of PSM morphogenesis and axis elongation in axolotl. (**a**) Gene expression of mesodermal (*brachyury*, blue) and neuronal (*sox2*, red) precursors in the posterior neural plate (NP) based on ISH [23]. (**b**) Topographic labelling of the left (shown here) or entire posterior NP in a white (*d/d*) neurula (stage 15) with a GFP^+^ posterior NP tissue graft from a transgenic donor. (**c**) Lateral aspect of a tailbud (stage 22); GFP^+^ graft-derived cells are either mesodermal (PSM), neuronal (NT) or dual-fated (NM). The neuronal cells can be identified via Sox2 immunostaining. Mesodermal cells were identified using GFP antibody (green). Further, they were localized in the PSM position of the dorsolateral trunk/tail after the neuronal cells had shifted to the posterior NT (see Figure 3). (**d**) Lateral aspect of tailbud (stage 28); GFP^+^ PSM cells (green) undergo an anterior turn and form a tissue strand that moves anteriorly while neuronal precursors (red) remain behind in the NT. Anterior PSM becomes segmented into somites. (**e**) Posterior end of larva (stage 35) where GFP^+^ PSM cells mix anteriorly with paraxial mesoderm derived from gastrulated mesoderm (grey). They form somites in the tail and posterior trunk. Sox2^+^ cells contribute to the posterior NT. (**f**) Summary of spatiotemporal order of PSM morphogenesis. Schematic of dorsolateral aspect of the left posterior end of a tailbud at the level of the NT (based on experiments shown in Figure 3, Figure 4, Figure 5, Figure 6 and Appendix A). Graft-derived NM progenitors in the PZ acquire either a neuronal fate (red, Sox2^+^) and become integrated into the posterior NT or develop into mesenchyme (green, GFP^+^) and undergo an anterior turn. After the turn cells first migrate away from the midline and then anteriorly forming the posterior and medial PSM. In the PZ, PSM cells are mainly oval and mediolaterally oriented, while posterior PSM cells are elongated and anteroposteriorly or mediolaterally-oriented. Medial and anterior PSM cells are mainly oval and anteroposteriorly or randomly oriented. Anterior PSM cells become elongated when forming somites and are then predominantly dorsoventrally oriented. These elongated cells get organized into somite rosettes [58]. PSM morphogenesis leads to an elongation of the posterior axis despite the mesenchyme for new somites being added in an anterior direction. (**g**) By transplanting small left median, paramedian or lateral GFP^+^ posterior NP grafts into white (*d/d*) hosts (see Figure 5), local patterns of PSM cell migration can be studied. (**h**) Summary of individual patterns of PSM cell migration on a horizontal section through the left posterior end of a larva at the level of the NT. Median-most cells migrate early to the furthest anterior position, close to the NT, while lateral cells reach only more posterior positions and are farther away from the NT laterally. This might correspond either to a delayed migration or a slower speed of the lateral-most cells. GFP^+^ cells derived from a paramedian position were found along the entire length of the PSM from its posterior region up to the anterior PSM. The individual substreams conduct general anterior movement (3 green arrows). The PSM cells display opposing gradients of cell mobility and density. In the forming somites, the cell density increases while cell mobility decreases (see columns in the right margin). Panels (**f**) and (**h**) summarize morphogenetic events at stages 28 to 35. Abbreviations: posterior NP: posterior neural plate; NT: neural tube; NM: neuro-mesodermal; PZ: posterior zone; PSM: presomitic mesoderm; a-p: anterior-posterior; d-v: dorso-ventral; m-l: mediolateral; ISH: *in situ* hybridization.

**Table 2 cells-12-01313-t002:** Number of GFP^+^ mesodermal and neural cells (average cell number per embryo, *n* = 3 embryos/stage) and the percentage of anteroposterior cell divisions among all cell divisions in the developing axolotl tail.

Stage	GFP^+^ Mesodermal Cells	GFP^+^ Neural Cells	Anteroposteriorly-Oriented Cell Divisions [%]
19	527	0	5.9
22	1095	27	1.4
28	1232	78	12.1
30	1989	1	4.4

**Table 3 cells-12-01313-t003:** Tail phenotypes after epidermis removal.

Epidermis Removal	Normal	Lateral Bending	Misshapen	Missing	*n* (Total Analysed Embryos)
Control	4 (80%)	0	0	1 (20%)	5
Left	2 (12.5%)	7 (43.8%)	4 (25%)	3 (18.8%)	16
Right	2 (20%)	2 (20%)	4 (40%)	2 (20%)	10
Bilateral	1 (10%)	2 (20%)	3 (30%)	4 (40%)	10

**Table 4 cells-12-01313-t004:** Tail phenotypes after epidermis removal combined with rescue treatments (embedding +/− fibronectin). Data presented as a number of control (ctrl) versus epidermis-deprived (epidermis^−^) embryos in each category (ctrl/epidermis^−^).

Condition	Normal	Deformed	*n* (Total)
3% methylcellulose	3/0	0/3	3/3
3% methylcellulose + fibronectin	3/0	0/3	3/3
1% agarose	3/0	0/3	3/3
1% agarose + fibronectin	2/0	1/3	3/3

## Data Availability

The data presented in this study are available in this article (including in the Appendix A).

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
