# Peer review of "The Role of Posterior Neural Plate-Derived Presomitic Mesoderm (PSM) in Trunk and Tail Muscle Formation and Axis Elongation"

_cells, 2023, doi:10.3390/cells12091313_

Round 1

Reviewer 1 Report

In this manuscript, the authors investigated a morphological analysis of axolotl, Ambystoma mexicanum, embryo. They focused on the posterior neural plate (NP)- derived presomitic mesoderm (PSM) and somites, which PSM gives rise to, in which they analyzed the behavior of each cell and the collective behavior of cells. In detail, the authors described the translocation of the cells in embryos (stages 22-40) using DiI labeling and GFP+ tissue grafts. They tracked the posterior NP-cells using GFP+ grafts and showed their location in the later stage by 3D reconstructions. They described the behavior of each GFP+ graft cell and analyzed the relationship to the surrounding cells. Finally, the authors examined the role of the epidermis in the formation of the PSM tissue strand by removing the epidermis.

The analyses by the authors are done soundly, and the data quality is adequate. Although most of their finding have already been found in other species, the data in this manuscript are informative and worthwhile to publish. However, a few concerns should be addressed.

1. Cell death or apoptosis in NP-cells and PSM cells may affect cell translocation or behavior. The author should check or discuss the cell death of apoptosis in the PSM.

2. In the analysis of cell shape and behavior (3.5.x, Fig 6, Fig7, and Fig S8), I don’t think that somite cells can be compared with other PSM cells because at least a part of somite cells are epithelized. The authors should discuss this point.

Minor points:

1. From lines 753 to 757, I think “Figure S9” should be “Figure S10,” if I don’t misunderstand.

2. In Figure 6d, the column for medial PSM is gray, but it should be the same color in Figure 6c for easier understanding.

Author Response

We would like to thank Reviewer 1 for a positive review of our study and helpful suggestions. We will address them below.

  1. Cell death or apoptosis in NP-cells and PSM cells may affect cell translocation or behavior. The author should check or discuss the cell death of apoptosis in the PSM.

Cell death occurs regularly during embryogenesis if not counteracted, for example during somite development in the chick [Sanders & Parker, 2002; doi: 10.1007/s004290100208]. This topic would have been also of interest in PSM cells of the developing axolotl if the focus had not been elsewhere. When grafting entire or left posterior NP tissue at stage 15 we were primarily interested in a qualitative study of an overall distribution and differentiation of neural plate (NP) and NP-derived PSM cells, as well as in a potential distinction between GFP+ Sox2- (mesenchyme) and Sox2+ cells (neurons) (see Figures 3-5). Given that apoptosis is a rare event in the developing PSM, at least in mouse [Borycki et al. 1999; doi: 10.1242/dev.126.8.1665], we focused on the overall morphogenesis and different cellular parameters. We do agree with the reviewer that a quantitative analysis of the PSM would also include a quantification of apoptotic/necrotic cells. This was, however, not the main interest of this study. Nonetheless, for a discussion of this interesting aspect, we have now added a following sentence to the manuscript (lines 573-576):

‘We have not assessed the effect of apoptosis or other types of cell death, however, previous studies in mouse PSM suggest that apoptosis in this tissue is limited and unlikely to have a major effect on morphogenesis [50].’

  1. In the analysis of cell shape and behavior (3.5.x, Fig 6, Fig7, and Fig S8), I don’t think that somite cells can be compared with other PSM cells because at least a part of somite cells are epithelized. The authors should discuss this point.

The major goal of analyzing the PSM in axolotl was to follow the morphology of these cells from PZ to somites. The somite as a final destination of PSM cells was included in the analysis to compare cell distribution and cell shape characteristics. The fact that PSM cells eventually take on a relatively long cell form which is ready to become arranged in an epithelial somite was, however, no reason to exclude these cells from the formal comparison.

Nonetheless, to clarify this point, we added the following sentences to the manuscript (lines 642-648):

‘In summary, in the developing PSM, elongated cells are more frequent in the posterior PSM than in other PSM regions. Most elongated cells could be found in the forming somites. These were previously reported to be highly elongated [30], [31], [34]. This is unlikely to be associated with motility but rather reflects cell shape change concomitant with epithelization as a result of mesenchyme-to-epithelial transition [34]. In contrast, the elongation of PSM cells before somite formation could be associated with their higher motility during tissue morphogenesis.’

Minor points:

  1. From lines 753 to 757, I think “Figure S9” should be “Figure S10,” if I don’t misunderstand.

We have corrected the Figure numbering from ‘Figure S9’ to ‘Figure S11’ (shifted because of addition of a new Figure S8, now lines: 780, 781, 783 and 785).

  1. In Figure 6d, the column for medial PSM is gray, but it should be the same color in Figure 6c for easier understanding.

We have corrected the colours in Figure 6d to match Figure 6c as recommended.

For reference please check the attached updated version of the manuscript including the supplementary data.

Reviewer 2 Report

Manuscript review 

cells –2330103 Stepein et al. 

The role of posterior neural plate-derived presomitic mesoderm (PSM) in trunk and tail muscle formation and axis elongation 

In this manuscript, Stepein et al. study the elongation of the axolotl embryo during tailbud stages when the mode of more classical gastrulation contributions changes to a different mode of size change. The authors extensively employ grafts of an axolotl transgenic strain expressing GFP or DiI lineage tracing. The grafted or labeled embryos were then analyzed by clearing, sectioning, and sometimes immunostaining. The authors reconstruct optical sections to analyze primarily the formation of the PSM originating in the tailbud region and its contribution to changes in embryo size, particularly elongation. The authors address the issues of cell proliferation, cell migration, multilayer contribution, division orientation, and others. The authors conclude that the PSM originating in the tailbud domain moves rostrally while the embryo grows posteriorly. 

This manuscript focuses on PSM originating in the tailbud and its developmental contribution to the embryo elongation. The authors center on the PSM of tailbud origin and in my review, I will refrain from mentioning again this fact for brevity. Overall, this manuscript includes some nice experiments and result analysis, but reaches some conclusions not supported by the data, lacks controls, or lacks experimental detail. The manuscript tries to address too many issues at the same time, making it hard to follow the resulting text that is exceedingly long and appears disorganized. For somebody decently knowledgeable in developmental biology of amphibian embryos and tailbud development, it became difficult to follow the multiple issues addressed, making the manuscript hard for the non-expert. Throughout the manuscript, the authors address the questions of; 

  • Rostral migration of the PSM. 

  • The developmental origin of the PSM. 

  • Cell proliferation within the migrating PSM. 

  • The coherent migration of the PSM. 

  • The dual specificity of the neuromesenchymal precursors. 

  • Volumetric vs non-volumetric development. 

I will address primarily the scientific issues of the manuscript although the manuscript should be shortened to remove superfluous text and edited for English, unclear sentences, and missing words. I will remark on some examples at the end of my review. 

Scientific comments 

1. Line 44 – There are other signaling pathways and mutations resulting in truncated rostro-caudal elongation of the embryo like BMP and RA. Collectively these types of mutations in mice were considered to induce “sirenomelia”, although the condition in humans is slightly different. 

2. Line 70 – There is an interesting paper (Rao 1994) and subsequent studies linking Brachyury also to neural differentiation. 

3. Section 3.1. - This section that aims at describing changes in size parameters as a function of developmental stage, based on embryo measurements, is relatively long due to a detailed description of A. mexicanum embryogenesis that could be replaced with a citation of a normal table of development. However, no mention of the numbers of embryos measured, standard deviation, or the number of biological replicates performed. 

4. Section 3.2 - The section begins with the aim to understand derivatives of the three germ layers, but we are shown only external pictures of the DiI injected embryos and the author’s conclusions, but no real histological and/or marker (immunostaining) evidence of the cells labeled and their fate. Are the pictures shown of the same embryo at different developmental stages? Importantly, this section also suffers from low sample sizes (n=5, Fig 2b) and there is no mention of how many times the experiment was repeated, i.e., reproducibility. Was the experiment performed once in one batch of embryos? 

This section also introduces for the first time the use of the GFP labeled grafts. There is no description of the transgenic line employed and why it is suitable. Also, many of the figures throughout the manuscript include beautiful schematic drawings but nowhere, the authors describe in detail how the grafts were prepared, how do they look, and what cell layers they include. As basically, this manuscript is a lineage analysis manuscript, this is critical information. 

5. Section 3.3 - This section appears unrelated to most of the manuscript. This section has a long introduction to the experiment (lines 354-363) but again suffers from a lack of important technical and experimental details. What do the grafts contain? The posterior part of the embryo or only the neural plate. This is one of the few instances where they use antibodies against Sox2 to label neural cells but unfortunately, they do not use any marker of the PSM which is the focus of their study.  

6. Section 3.4 - Why two sections to describe one experiment? It just increases the text. Very few neural cells are marked as coming from the graft but they do not appear to be Sox2 positive. Also, it is very disconcerting that some pictures are actual immunostainings (3b-d) while the neural tube in 3e is pseudo-colored similarly. Same comments as to Section 3.3. 

7. Sections 3.4.1-3.4.2 - Again, multiple sections and extensive text to describe the 3D reconstructions when it should be done more concisely. As before, serious lack of markers or good histological sections to identify the differentiation fates of the grafted cells. In 3.4.1 the authors discuss, contralateral migration, cell cohort volumes, and volumetric vs non-volumetric changes. Issues of volume would be clearer next to the cell proliferation studies and not while the fate is still being discussed. 

8. Table 1 – This table is confusing as it aims at almost summarizing most of the manuscript, but different lines have text while others have numbers. 

9. Section 3.5 - In this section the authors count the GFP+ cells in the optical sections to monitor cell numbers. Also here, neural cells are Sox2+ but PSM is Sox2- and not by a positive marker. 

This section also includes an analysis of oriented cell division based on vibratome sections that are not shown. In Table 2, is the % of anteroposteriorly oriented cell divisions out of all cells or only the mitotic cells? 

10. The authors use multiple times the term “manually segmented” throughout the manuscript. This term should be explained as it does not relate to the embryonic process of segmentation or dissection of the embryo. 

11. Section 3.6 - The authors describe the epidermis removal + graft embryos, what were the controls? What happens to control embryos with epidermis removal only? Is there a phenotype? Why is a limited Table provided as the summary of the results when there is a good Supplemental figure that could incorporate this data that should also include statistical analysis in the graphs? 

The same comments apply to the results summarized in Table 4 and the corresponding Supplemental figure. 

Some text comments 

1. Line 187 – Missing word, anteroposterior axis. 

2. Line 359 – should be “existence” 

3. Line 411– should be “consistent” 

4. Line 719 - Missing word, at which point? 

Author Response

We would like to thank Reviewer 2 for a critical review of our study and many helpful suggestions, which are addressed below.

We have edited the manuscript with the help of a native English speaker in order to remove unclear sentences, grammatical and spelling errors. We also shortened some paragraphs according to the reviewer’s suggestions.

Scientific comments 

  1. Line 44 – There are other signaling pathways and mutations resulting in truncated rostro-caudal elongation of the embryo like BMP and RA. Collectively these types of mutations in mice were considered to induce “sirenomelia”, although the condition in humans is slightly different.

As the reviewer rightly pointed out, the signalling pathway mutations listed in the Introduction are not the only ones reported to affect posterior body elongation. They were listed merely to provide examples of such signalling. According with the reviewer’s suggestion we now modified the text to also include the BMP and RA pathways (lines 45-48):

‘Critically, the interruption of the formation of the posterior body, e.g., by mutations of Brachyury, Wnt, FGF, BMP or RA pathway components, leads to a truncated rostro-caudal elongation of the embryo that largely affect the morphogenesis of the anterior trunk and head [3], [6] that is also known as “sirenomelia” [9]’.

A new citation [9] was added: Garrido-Allepuz et al., 2011, doi: 10.1242/dmm.007732

  1. Line 70 – There is an interesting paper (Rao 1994) and subsequent studies linking Brachyury also to neural differentiation.

We thank the reviewer for this suggestion. The reference to Rao (1994) was added as [28] to the text along with the following sentence (lines 72-74):

‘Interestingly, a truncated Xenopus Xbra mutant that has lost the mesodermalizing activity can lead to the formation of neural structures in animal cap explants [Rao, 1994; DOI: 10.1101/gad.8.8.939].’

  1. Section 3.1. - This section that aims at describing changes in size parameters as a function of developmental stage, based on embryo measurements, is relatively long due to a detailed description of A. mexicanum embryogenesis that could be replaced with a citation of a normal table of development. However, no mention of the numbers of embryos measured, standard deviation, or the number of biological replicates performed.

Section 3.1 (relating to Figure 1a-i) is based entirely on a re-analysis of the published data from [48], which does not report the statistical details. However, we would insist that our description is retained. The description of stages provided in our study is more detailed in comparison to the original table [48]. It also includes own photographs of actual embryos (Figure 1a-g) and an analysis of changes in embryonic dimensions (Figure 1h-i). It is a valuable addition, especially for readers not familiar with the axolotl. Nonetheless, we have followed the reviewer’s suggestion and shortened the text as follows (now lines 185-201):

‘To facilitate a detailed investigation of the morphogenetic movements of posterior NP-derived PSM we first analyzed the shape changes occurring in the axolotl embryo between a neurula (stage 14) and a prehatching larva (stage 40; Figure 1), based on the data from [48]. As these descriptions are too short, we have to go into more details here. Stages from a fertilized egg to a late gastrula have a constant diameter of 2 mm and are spherical. From an early neurula (stage 14; Figure 1a) onwards embryos become elliptical with a progressively longer anteroposterior (AP) axis. The neural folds become elevated, approach each other towards the dorsal midline (middle neurula, stage 16; Figure 1b) and fuse (stage 19; Figure 1c). At stage 22, an early tailbud stage has developed with fused neural folds and a prospective head, trunk and tail (Figure 1d, n-o). The embryo elongates along the AP axis through stages 28 (Figure 1e) and 30 (Figure 1f). At stage 35 a larva with a stretched AP axis and an extended tail is formed (not shown) that continues to lengthen while head, trunk and tail structures become more clearly discernible. The pre-hatched larva (stage 40; Figure 1g) has an elongated body with a central NT and a notochord in the trunk and tail, flanked bilaterally by paraxial mesoderm and is covered with epidermis (Figure 1j-m). The tail differs from the trunk by lacking LPM and endoderm [7], [8].’

  1. Section 3.2 - The section begins with the aim to understand derivatives of the three germ layers, but we are shown only external pictures of the DiI injected embryos and the author’s conclusions, but no real histological and/or marker (immunostaining) evidence of the cells labeled and their fate. Are the pictures shown of the same embryo at different developmental stages? Importantly, this section also suffers from low sample sizes (n=5, Fig 2b) and there is no mention of how many times the experiment was repeated, i.e., reproducibility. Was the experiment performed once in one batch of embryos?

This section also introduces for the first time the use of the GFP labeled grafts. There is no description of the transgenic line employed and why it is suitable. Also, many of the figures throughout the manuscript include beautiful schematic drawings but nowhere, the authors describe in detail how the grafts were prepared, how do they look, and what cell layers they include. As basically, this manuscript is a lineage analysis manuscript, this is critical information.

To address the reviewer’s question regarding the identity of the pictured embryos: the pictures presented in Figure 2c, as well as in Figures S5-6, show the same embryo in each row. The course of labelling has been followed for each animal from stage 22 to stage 40. The location of DiI in specific tissue structures can be assessed visually based on the knowledge of embryonic morphology and does not require additional staining. The thickness of the tissues (endoderm, LPM, epidermis) can be assessed in Figure 1j-o. Moreover, immunostaining at intermediate timepoints is not possible because the embryos have to remain alive till stage 40.

In respect to the low sample size, we would like to point out to the reviewer that DiI-labelling and GFP+ tissue grafting experiments are technically very challenging. Some of the injected/grafted embryos fail to develop properly which leads to a dropout rate of around 50%. Therefore, it is not practically feasible, to still analyze larger animal numbers than done here. Animal welfare (3R rules) also restricts the maximal number of animals that can be used to a minimum required to reach a valid conclusion. The standard number of animals used for analysis in such experiments is typically three; we already went beyond that and we do not think a further increase in animal numbers would yield any important changes or additional conclusions. Each of these experiments was also performed on multiple batches of embryos.

Regarding questions to the use of the transgenic line employed to generate the GFP labeled grafts we refer the reviewer to the Supplementary Materials and Methods, as quoted below:

1. Animals

Eggs of white mutant (d/d) (host) and transgenic GFP-expressing white mutant (d/d) (donor) females (CAGGs:EGFP; designated as GFP+) of the Mexican axolotl (Ambystoma mexicanum) were obtained from the axolotl-facility at the Center for Regenerative Therapies (CRTD), TU Dresden.’

Similarly, the detailed description of the grafting procedure can be found in the Supplementary Materials and Methods (3. Homotopic grafting of GFP+ tissues). The posterior neural plate grafting procedure is also identical to the already published protocol quoted throughout the text (references [23] and [24]:  Taniguchi et al., 2017; doi: 10.1016/j.ydbio.2016.12.023; and Taniguchi et al., 2015; doi: 10.1002/9783527808465.EMC2016.6574, respectively). The referred publications provide a detailed description of the grafts, which in our view makes the repetition of such description in the current manuscript unnecessary.

  1. Section 3.3 - This section appears unrelated to most of the manuscript. This section has a long introduction to the experiment (lines 354-363) but again suffers from a lack of important technical and experimental details. What do the grafts contain? The posterior part of the embryo or only the neural plate. This is one of the few instances where they use antibodies against Sox2 to label neural cells but unfortunately, they do not use any marker of the PSM which is the focus of their study.

Regarding the identity of the grafted tissue, we would like to again refer the reviewer to the Supplementary Materials and Methods (3. Homotopic grafting of GFP+ tissues).  In brief, as stated in lines 360-368, the GFP+ plate graft contains the posterior-most fifth of the neural plate, or its parts (only left half or smaller median, paramedian or lateral regions), which consists of prospective PSM and NT progenitors, as was already described in the cited publications (references [23] and [23]:  Taniguchi et al., 2017; doi: 10.1016/j.ydbio.2016.12.023; and Taniguchi et al., 2015; doi: 10.1002/9783527808465.EMC2016.6574, respectively). The sizes of grafts used in each experiment were also described throughout the text, next to respective experiments.

We admit that it would be advantageous to stain the PSM tissue with a specific marker. However, no antibody is available that specifically stains the PSM in axolotl. Therefore, throughout this study we identify PSM cells via positive GFP staining combined with the lack of neuronal Sox2 staining. Based on established identity of the descendants of the grafted posterior neural plate (references [23] and [23]:  Taniguchi et al., 2017; doi: 10.1016/j.ydbio.2016.12.023; and Taniguchi et al., 2015; doi: 10.1002/9783527808465.EMC2016.6574, respectively) this method is suitable to identify PSM cells. In order to clarify this issue to potential readers we have now added the following sentences in the text (lines 373-377):

‘Due to the lack of a mesenchyme-specific marker, which makes the labelling of the PSM cells in the axolotl impossible, we combined GFP staining (which marks all cells derived from the posterior NP graft) with Sox2 neuronal staining, in addition to DAPI staining of the nuclei. This allows an identification of the PSM cells as GFP+ and Sox2-.’

  1. Section 3.4 - Why two sections to describe one experiment? It just increases the text. Very few neural cells are marked as coming from the graft but they do not appear to be Sox2 positive. Also, it is very disconcerting that some pictures are actual immunostainings (3b-d) while the neural tube in 3e is pseudo-colored similarly. Same comments as to Section 3.3.

In accordance with the reviewer’s suggestions, we now shortened and rewrote Section 3.3 to reduce the text.  Section 3.4 consists of two parts. In section 3.4.1 we grafted the left posterior fifth of the neural plate, while in section 3.4.2 we grafted median, paramedian or lateral subregions of the posterior plate (as described in detail in the Supplementary Materials and Methods (3. Homotopic grafting of GFP+ tissues) as well as in the respective sections in the main manuscript. These experiments analyze different aspects of PSM morphogenesis and therefore are kept separate.

Regarding the neuronal cells derived from the graft: in general, there are very few GFP+ graft-derived neuronal cells present. This can be seen in the quantification in Table 2, which shows that only between 0 and 6% of the identified GFP+ cells are Sox2+. This shows that although the posterior neural plate of the axolotl consists of dual-fated NM progenitors, the vast majority of the cells are mesenchymal.

The Sox2 staining is indeed not present in Figure 3e and therefore the neural tube is only indicated with a red pseudocolor. The reason for this is that this Figure only serves as proof of principle for the 3D reconstructions of PSM morphogenesis at different developmental stages, in which all GFP+ cells can be identified based on the immunostaining in optically-cleared whole mounts of axolotl embryos. For all other experiments the Sox2 staining was included (as shown in Figure 3c-d, Figure 4b and Figure 5) and used to identify neuronal cells in the neural tube. We added a sentence to the manuscript to clarify the difference between Figure 3c-d and Figure 3e (lines 390-394):

‘To obtain a wholistic picture of PSM morphogenesis, in addition to studying GFP+ PSM development on vibratome sections and whole mounts as above (Figure 3b-d), we decided to analyze this system also in 3D tissue reconstructions of the tail region in whole mounts of axolotl embryos. In these reconstructions individual GFP+ cells are segmented and analyzed (Figure 3e).’

  1. Sections 3.4.1-3.4.2 - Again, multiple sections and extensive text to describe the 3D reconstructions when it should be done more concisely. As before, serious lack of markers or good histological sections to identify the differentiation fates of the grafted cells. In 3.4.1 the authors discuss, contralateral migration, cell cohort volumes, and volumetric vs non-volumetric changes. Issues of volume would be clearer next to the cell proliferation studies and not while the fate is still being discussed.

As mentioned before (in the answer to point 5) we do not have a marker for mesenchyme in axolotl embryos. We do however use Sox2 as a marker for neuronal cells in the NT (Figure 3c-d, 4b and 5, red). The use of histological sections makes no sense here because all labelled cells can already be seen in 3D in the optically-cleared whole mount tissue.

While we understand that it may be confusing, the inclusion of both the analysis of tissue volume and of individual cell volumes is important in this section because here the parameters underlying the changes in tissue dimensions are being addressed. Cell volume changes are included in this section specifically to check whether they can explain the overall tissue volume changes.

The cell fate is not discussed in this section, instead it is discussed earlier (Section 3.3 Dual cell fate of the posterior NP progenitors).

  1. Table 1 – This table is confusing as it aims at almost summarizing most of the manuscript, but different lines have text while others have numbers.

Table 1 is indeed thought to serve as an easy overview/summary for most of the results in this study that otherwise might get lost in the text. It should therefore help the reader to quickly find relevant information. As the data consist of both descriptive and quantitative parameters, we necessarily used both numerical and verbal indications.

  1. Section 3.5 - In this section the authors count the GFP+ cells in the optical sections to monitor cell numbers. Also here, neural cells are Sox2+ but PSM is Sox2- and not by a positive marker. This section also includes an analysis of oriented cell division based on vibratome sections that are not shown. In Table 2, is the % of anteroposteriorly oriented cell divisions out of all cells or only the mitotic cells?

As mentioned before (in the answer to point 5 and 7) we do not have a marker for the PSM mesenchyme in axolotl embryos.

An example image of the vibratome section showing a dividing cell in anaphase stained with propidium iodide (marked with an arrow) is now added as Figure S8 together with a corresponding supplementary figure legend:

‘The image shows a single plane of a 3D reconstruction of a z-stack through a vibratome section of the PSM of a stage 22 embryo. DNA was stained with propidium iodide. The magenta dots mark the opposing poles of the nucleus (labelled A and B) connected by a solid white line, which indicates the orientation of cell division. The division plane (dashed white line) is perpendicular to this line. The cell nuclei of surrounding cells are not in mitosis.’

In addition, a sentence referring to the new Figure S8 was added (lines 582-585):

‘To test if oriented cell divisions play a role in anteroposterior PSM expansion, individual mitotic cells were identified based on the nuclear DNA appearance in thick vibratome sections stained with propidium iodide (for a representative image see Figure S8).’

The percentages of anteroposteriorly oriented cell divisions reported in Table 2 were calculated based only on mitotic cells. This clarification was also added to the Table 2 caption (lines 577-579):

‘Number of GFP+ mesodermal and neural cells (average cell number per embryo, n = 3 embryos/stage) and the percentage of anteroposterior cell divisions among all cell divisions in the developing axolotl tail.’

  1. The authors use multiple times the term “manually segmented” throughout the manuscript. This term should be explained as it does not relate to the embryonic process of segmentation or dissection of the embryo.

Manual segmentation is described in the Supplementary Materials and Methods under the Section 7.2 and shown in Figure S2 (with a description in the legend):

‘7.2 Manual cell segmentation: Individual mesodermal cells within the tail were analysed for different cellular parameters. To measure these parameters, vibratome sections through embryos containing small median posterior neural fold grafts from lifeact-GFP transgenic embryos (2.3.3.) were imaged as a series of z-stacks (Figure S2a). For single cell analysis, all cells which were visually distinguishable in all dimensions from other GFP-labelled cells and contained a nucleus in the imaged z-stack were manually segmented with the Fiji Plugin Segmentation editor (Figure S2b). Thereby, cell borders of each cell were manually outlined in all images of a z-stack. The segmented cells were then used for ellipsoid-fitting and calculation of cellular parameters.’

  1. Section 3.6 - The authors describe the epidermis removal + graft embryos, what were the controls? What happens to control embryos with epidermis removal only? Is there a phenotype? Why is a limited Table provided as the summary of the results when there is a good Supplemental figure that could incorporate this data that should also include statistical analysis in the graphs? The same comments apply to the results summarized in Table 4 and the corresponding Supplemental figure.

In the epidermis removal experiments summarized in Table 3 all embryos were generated by grafting the left half of the GFP+ posterior neural plate tissue.  The embryos were then allowed to develop until stage 22, at which stage an ipsi-, contra- or bilateral epidermis removal was performed (experimental embryos) or no epidermis removal was done (control embryos). The control GFP+ graft-containing embryos without epidermis removal only rarely show a phenotype, as indicated in Table 3. We did not perform epidermis removal in embryos without grafting as that would not be an appropriate control for this experimental setup. We now clarify this point in lines 747-748:

‘Chimeric embryos with the left GFP+ posterior neural plate grafts without epidermis removal were used as controls.’

We would like to point out that the quantification of the observed phenotypes for the epidermis removal experiments is indeed provided as the Figure S10, including the statistical analysis in this supplementary figure legend.

Regarding rescue experiments after epidermis removal summarized in Table 4, both experimental and control embryos were NOT generated by grafting to minimize the effects of the grafting procedure (the relatively small effect seen for control embryos in Table 3). Here control embryos refer to embryos in which no epidermis removal was performed and which were embedded in either media with or without fibronectin and compared to experimental embryos with epidermis removal embedded in the same conditions. We incorporated this information clearly in the text (lines 813-814):

‘Stage 22 embryos in which no epidermis removal was performed embedded in the same conditions served as controls.’

As requested by the reviewer the summary of this experiment is now provided in Figure S12 (with a corresponding figure legend). In addition, sentences referring to the new Figure were added (lines 810-813):

‘The epidermis was removed on the left side of embryos at stage 22 and embryos were embedded in either soft 3% methylcellulose or a stiffer 1% agarose gel to manipulate the stiffness of the physical barrier (Figure S12a).’

and lines 827-828:

‘Representative images of control and epidermis-removed embryos embedded in different conditions are provided in Figure S12b.’

Some text comments 

  1. Line 187 – Missing word, anteroposterior axis. 
  2. Line 359 – should be “existence”
  3. Line 411– should be “consistent” 
  4. Line 719 - Missing word, at which point?

We corrected the spelling and grammatical mistakes throughout the text according to the reviewer’s suggestions.

For reference please check the attached updated version of the manuscript including the supplementary data.

Reviewer 3 Report

In the paper by Stepien et al., the authors analyzed the morphogenesis of several tissue types during the elongation of the axolotl body axis. The authors took advantage of classic embryology approaches, in combination with sophisticated imaging methods to investigate the behavior of different tissues (presomitic mesoderm, epidermis, lateral plate mesoderm, and endoderm) and cells during the elongation of the body axis, focusing on the migration of cells during the extension of the posterior presomitic mesoderm. The experiments were done by tracing DiI-labeled cells or GFP+ tissue grafts during the elongation of the A/P axis and 3D reconstructions. The results show that during the elongation of the body axis, presomitic mesoderm, lateral plate mesoderm, and endoderm all shift anteriorly. The PSM makes an important contribution to the axis elongation in the posterior body. Both NT and notochord play an important role in directing the PSM tissue flow. The epidermis acts as the lateral limit to PSM cell spreading during the elongation. As a result, the PSM cells are directed anteriorly toward the site of somite formation. Meanwhile, the PSM narrows down, leading to the posterior elongation of the embryo. Interestingly, the migrating mesenchymal cells undergo changes in collective cell behavior, although less densely packed cells are present. Overall, I feel this work was done elegantly. The authors were very thorough and carefully assessed the migratory behavior of cells during morphogenesis. I very much enjoy reading the paper. 

That said, this work is highly descriptive. While the authors have done a very nice job describing how cells migrate during the elongation of the body axis, the molecular mechanisms governing the migratory behavior are not investigated. Therefore, I would recommend the authors publish this work in a more specialized journal. Listed below are some minor suggestions. 

The authors stated that the embryo grows volumetrically during late pre-larval and early larval stages. This conclusion would be more solid if the authors could actually measure the overall volume of the embryo.

In Fig2, the authors use Drel to indicate the relative distance to the anterior body border. Since the head develops continuously from stage 22 to stage 40, it seems better to use the location of the first somite as the marker for the measurement. 

Author Response

We would like to thank Reviewer 3 for a positive review of our study and thoughtful suggestions. We will address them below.

1. The authors stated that the embryo grows volumetrically during late pre-larval and early larval stages. This conclusion would be more solid if the authors could actually measure the overall volume of the embryo.

The analysis presented in Figure 1h-i is based on the data contained in the classical developmental table [48]. We agree with the reviewer 3 that to gain more insight into volumetric changes of the entire embryo a more thorough analysis of newly generated data would be needed. However, the primary focus of this study is on the morphogenesis of the posterior body PSM, hence such analysis was beyond its scope. In fact, we performed detailed measurement of the PSM dimensions, including the volume, as presented in Figure 4c-g.

In Fig2, the authors use Drel to indicate the relative distance to the anterior body border. Since the head develops continuously from stage 22 to stage 40, it seems better to use the location of the first somite as the marker for the measurement.

The reviewer is right to point out the potential differences in analysis depending on whether the label position is normalized to an entire embryonic length including the head or the length starting from the 1st somite. We have conducted measurements of the head region at stages 22, 30, 35 and 40 and the head to body length ratio is around 13%, 19%, 21% and 18%, respectively. That accounts for at mostly only around 8% difference between stages 22 and 35 and even lower differences between other stages, suggesting that using an alternative reference system would not significantly change our conclusions.

Round 2

Reviewer 1 Report

The authors addressed the concerns. 

Reviewer 3 Report

It appears that none of my concerns have been addressed. As stated in my previous review, while the authors have done a very nice job describing cell migratory behaviors during morphogenesis, the underlying molecular mechanisms are not investigated. I still think the authors should publish this work in a more specialized journal. 

As to the other two minor points I raised previously, I still think the authors should measure the overall volume of the embryo during different stages. Since they keep emphasizing volumetric vs non-volumetric growth in the paper, it is important to measure the volume of the embryo. Measuring dimension is not sufficient. If the authors don't want to measure the volume of the embryo, they should avoid talking about the volumetric or non-volumetric growth. As to using the first somite as the marker, I agree with the author, the result will likely be very similar to the one shown in fig 2. I just think it would be more accurate to measure based on the location of the somite.